# Lifespan prolonging mechanisms and insulin upregulation without fat accumulation in long-lived reproductives of a higher termite

Sarah Séité[1,2,15], Mark C. Harrison [3,15], David Sillam-Dussès[4], Roland Lupoli[1,2], Tom J. M. Van Dooren[5,6], Alain Robert[4], Laure-Anne Poissonnier[7], Arnaud Lemainque[8], David Renault[9,10], Sébastien Acket[11], Muriel Andrieu[12], José Viscarra[13], Hei Sook Sul[13], Z. Wilhelm de Beer[7], Erich Bornberg-Bauer[3] & Mireille Vasseur-Cognet [1,2,14 ✉]

Kings and queens of eusocial termites can live for decades, while queens sustain a nearly maximal fertility. To investigate the molecular mechanisms underlying their long lifespan, we carried out transcriptomics, lipidomics and metabolomics in *Macrotermes natalensis* on sterile short-lived workers, long-lived kings and five stages spanning twenty years of adult queen maturation. Reproductives share gene expression differences from workers in agreement with a reduction of several aging-related processes, involving upregulation of DNA damage repair and mitochondrial functions. Anti-oxidant gene expression is downregulated, while peroxidability of membranes in queens decreases. Against expectations, we observed an upregulated gene expression in fat bodies of reproductives of several components of the IIS pathway, including an insulin-like peptide, *Ilp9*. This pattern does not lead to deleterious fat storage in physogastric queens, while simple sugars dominate in their hemolymph and large amounts of resources are allocated towards oogenesis. Our findings support the notion that all processes causing aging need to be addressed simultaneously in order to prevent it.

[1] UMR IRD 242, UPEC, CNRS 7618, UPMC 113, INRAe 1392, Paris 7 113, Institute of Ecology and Environmental Sciences of Paris, Bondy, France. [2] University of Paris-Est, Créteil, France. [3] Institute for Evolution and Biodiversity, University of Münster, Münster, Germany. [4] University Sorbonne Paris Nord, Laboratory of Experimental and Comparative Ethology, UR4443 Villetaneuse, France. [5] UMR UPMC 113, IRD 242, UPEC, CNRS 7618, INRA 1392, PARIS 7 113, Institute of Ecology and Environmental Sciences of Paris, Paris, France. [6] Naturalis Biodiversity Center, Leiden, The Netherlands. [7] Department of Biochemistry, Genetics and Microbiology, Forestry and Agricultural Biotechnology Institute, University of Pretoria, Pretoria, South Africa. [8] Genoscope, François-Jacob Institute of Biology, Alternative Energies and Atomic Energy Commission, University of Paris-Saclay, Evry, France. [9] University of Rennes, CNRS, ECOBIO (Ecosystems, biodiversity, evolution) - UMR, 6553 Rennes, France. [10] University Institute of France, Paris, France. [11] University of Technology of Compiègne, UPJV, UMR CNRS 7025, Enzyme and Cell Engineering, Royallieu research Center, Compiègne, France. [12] Cochin Institute, UMR INSERM U1016, CNRS 8104, University of Paris Descartes, CYBIO Platform, Paris, France. [13] Department of Nutritional Sciences and Toxicology, University of California, Berkeley, CA, USA. [14] INSERM, Paris, France. [15] These authors contributed equally: Sarah Séité, Mark C. Harrison. ✉email: mireille.vasseur@inserm.fr

Aging affects almost all living organisms. It is characterized by the decay of several cellular and physiological functions, such as a deleterious accumulation of lipids[1,2], DNA damage[3], or a reduction in mitochondrial functioning which exacerbates oxidative stress[3,4]. In most multicellular organisms, surgical or genetic interventions that reduce fecundity increase lifespan[5], suggesting that fecundity and longevity are negatively correlated[6,7]. This pattern is usually explained by a trade-off, where resources allocated to fecundity are no longer available for somatic maintenance and thus longevity[6,8,9]. The regulatory mechanisms and signaling pathways controlling the allocation of resources in this trade-off remain insufficiently understood[10]. The main molecular theories of aging propose different purposes for the allocation of resources to somatic maintenance, such as controlling damage accumulation, preserving mitochondrial functioning, or avoiding the deleterious accumulation of resources caused by inappropriate (e.g., insulin) signaling[3].

Our understanding of the causes of aging stems to a large extent from studies on short-lived model organisms[11,12]. Eusocial insects such as termites, ants and some bees and wasps seem to defy the trade-off between reproduction and longevity[13–16]. In the fungus-growing[17] termite *Macrotermes natalensis* (Termitidae, Blattodea), individuals differentiate by irreversible developmental plasticity into six distinct castes (major male and minor female workers, major and minor female soldiers, queens, and kings[17,18]). Queens and kings (reproductives) are for more than 20 years[13,15] confined to a royal cell where they mate regularly[19] and exhibit an extraordinarily long lifespan, while the median lifespan among sterile workers is 56 days[20,21]. Long-lived physogastric queens (i.e., with a hypertrophic abdomen) lay thousands of fertile eggs per day[22] and achieve close to their maximum possible fertility for a prolonged time without apparent signs of aging and thus with a negligible cost of reproduction[23]. Recently, transcriptomic studies in different taxa of social insects have proposed that downstream components of the Insulin/insulin-like growth factor (IGF-1) signaling (IIS) and the target of rapamycin (TOR) pathways seem essential for bypassing the fecundity/longevity trade-off[16,24–27]. This could be of broad relevance, since the dysregulation of these nutrient-sensing IIS or TOR pathways leads to the impairment of lipid metabolism and the development of metabolic, age-related pathologies, such as type 2 diabetes and insulin resistance, in a wide range of organisms from *Drosophila* to humans[28,29]. Metabolomic and lipidomic studies in social insects are now required to couple the detected gene expression plasticity between castes to the differential allocation of resources to reproduction and to prolonged healthy lifespan. In termite queens which remain fertile for many years, most major aging processes seem arrested[23]. In long-lived termite kings, one expects the same, but the trade-off might need less bypassing if the reproductive investment is systematically lower than in queens.

We investigated the molecular and physiological mechanisms allowing mature *M. natalensis* reproductives to maximize longevity, simultaneously with a large and long-term reproductive effort. To address this question, we performed tripartite -omics analyses (transcriptomics, lipidomics, and metabolomics) on queens, kings, and adult workers of these highly social termites. We carried out metabolomic analyses on hemolymph, and concentrated the transcriptomic and lipidomic analyses on the abdominal fat body. In insects, the fat body is central for intermediary metabolism and energy balance[2,30,31]. Moreover, it has been shown in *Drosophila* that a reduction of IIS or TOR pathway activity in this tissue can extend lifespan substantially while reducing female fecundity[32–34]. We compared gene expression between reproductives and workers of 20-year-old natural colonies. To understand their enormous developmental plasticity, we

complemented this with analyses on three stages of queen reproductive maturation in laboratory colonies. By corroborating and comparing these transcriptomic findings with our metabolic and lipidomic results, we were able to gain valuable insights into age- and caste-specific differences in the fat body metabolism linked to fertility and longevity.

We expected gene expression to differ between reproductives and sterile castes in downstream components and targets of the IIS and TOR pathways, for which caste bias has been detected across other social insect taxa[16], as well as patterns specific to our model system. In addition, we expected expression patterns in reproductives indicative of lifespan-prolonging mechanisms, of bypassing the reproduction/lifespan trade-off in queens in particular, and confirmation of these processes in the metabolomic and lipidomic analyses. We observed an upregulation of genes for different lifespan-prolonging mechanisms in mature queens and kings, supporting a robust mitochondrial functioning and increasing genome stability. An anticipated downregulation of the TORC1 signaling occurs in long-lived reproductives, but in contrast to previous studies, we observe an unexpected 800-fold upregulation of an insulin-like peptide gene which we called *Ilp9*. We correlate this phenomenon to a non-canonical downregulation of *midway* (*mdy*) involved in triglyceride lipid synthesis. The upregulation of the *Ilp9* gene coincides with high glucose and surprisingly low trehalose levels in the hemolymph of mature queens. The apparent insulin increase is associated with an upregulation of specific gene programs involved in the synthesis of proteins and specific lipids with low oxidation potential, destined for oogenesis rather than fat storage, thus involved in bypassing the fecundity/longevity trade-off. Consistent with this, lipidomic analyses demonstrate low concentrations of preferentially stored lipids (triglycerides) in the fat bodies of mature queens and increased concentrations of lipids destined for oogenesis (diglycerides).

## Results

**Specific regions of the gene co-expression network are activated in the fat bodies of different castes and during queen maturation.** To investigate caste-specific differences between reproductives and workers in natural conditions, *M. natalensis* queens (QT4) and kings (KT4), which were over 20-years old, and short-lived female (minor[35]) workers (FW) were sampled in field colonies ("Methods" and Supplementary Table 1). To study the dramatic adult developmental plasticity shown by *M. natalensis* termite queens in their reproductive system and fat body during maturation, we carried out a longitudinal study in laboratory colonies established from imagoes collected from the same field colonies (QT0, virgin queens) ("Methods" and Fig. 1). Incipient colonies were each founded from one male and one female imago and raised for 31 months following a protocol based on the natural life history of *Macrotermes* species[20,36] ("Methods" and Fig. 1). Queens of laboratory colonies were sampled at 3 months after establishment (QT1), 9 months (QT2), and 31 months of age (QT3). To investigate differential gene expression between our samples, we analyzed a total of 25 transcriptomes of abdominal fat bodies. RNA-sequencing data showed that variation in expression between castes and queen stages was greater than between colonies (Principal Component Analysis, PCA of top 500 genes in terms of variance, Supplementary Fig. 1a). Removing FW expression from the PCA allowed a clearer separation of the reproductive individuals along the first two axes (Supplementary Fig. 1b).

We then carried out a signed, weighted gene co-expression network analysis[37]. The resulting gene co-expression network (GCN) allowed us to identify nine modules of particularly

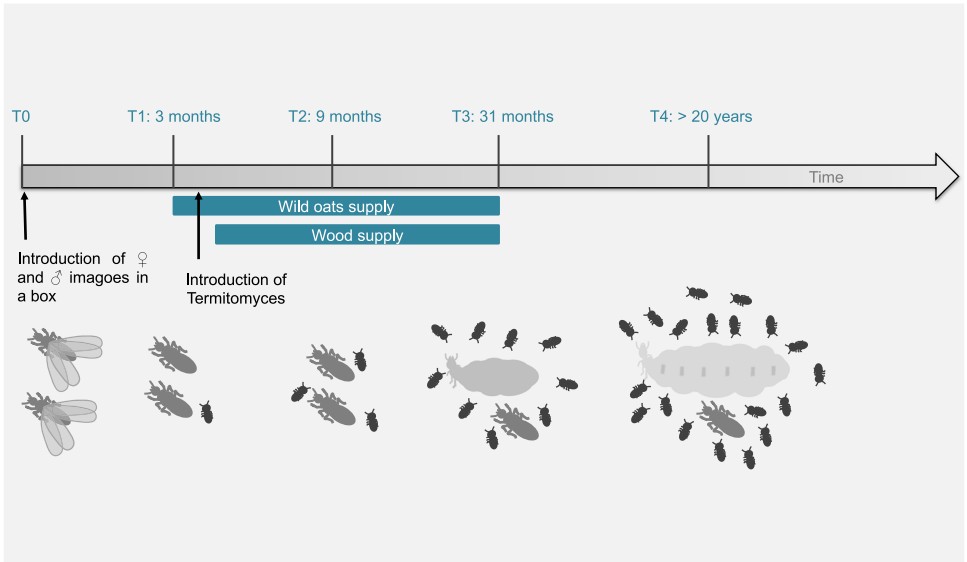

**Fig. 1 Overview of the model system.** Timeline of the different stages of *Macrotermes natalensis* colonies founded from one male and one female imago each (T0). Queens from incipient laboratory colonies were sampled at 3 months after colony establishment (QT1), 9 months (QT2), and 31 months (QT3). Field termite colonies over 20-years old are added. From field colonies, queens (QT0 and QT4), workers (FW), and kings (KT4) were sampled. Wild oats were supplied to the laboratory colonies from 3 months onward and the fungus *Termitomyces* sp. was introduced artificially in 3.5-month-old colonies. Wood was supplied from 4.5 months onward. The drawings represent winged imagoes, workers, kings, and queens at the different stages (physogastric queens become larger). Replication and sampling in our incipient colonies are further described in Supplementary Table 1.

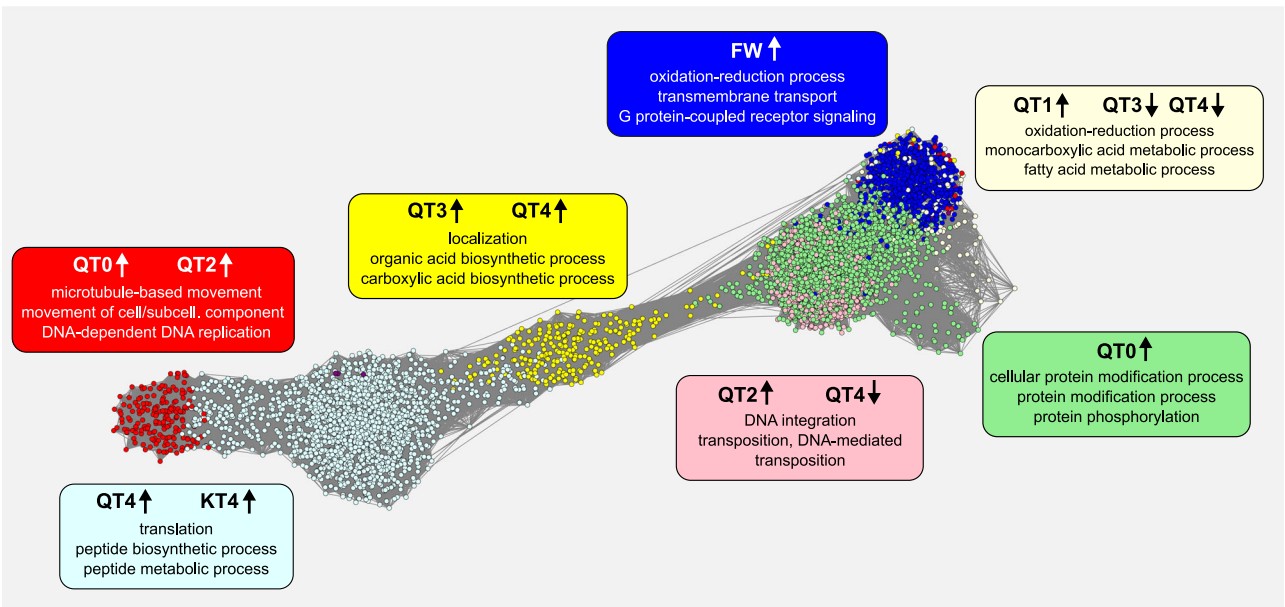

**Fig. 2 Weighted gene co-expression network analysis (WGCNA).** The gene co-expression network is displayed with genes as nodes and edges representing the co-expression relationships between genes. Nine modules were detected within which gene expression was especially strongly correlated (see Supplementary Fig. 2 for more details.) Shown are the seven largest modules, represented by colors. For each module, information is supplied on the caste or queen stage with which module expression is significantly correlated; upward arrow indicates a significant increase, downward arrow a significant decrease, in expression. The top three enriched GO terms associated with the gene members of each module are displayed (for full lists of significant GO terms, see Supplementary Data 1). This network was created with Cytoscape (version 3.8.0; Shannon et al.[77]) on a reduced representation of the WGCN containing the top connected genes (see "Methods" for more details).

strongly co-expressed genes associated with castes or with queen ages (Fig. 2, Supplementary Fig. 2, and Supplementary Data 1). In accordance with the PCA (Supplementary Fig. 1a), we found modules strongly and uniquely correlated with FW (blue module, 1103 genes; Supplementary Fig. 2 and Supplementary Data 1) and KT4 castes (plum1 module, 116 genes; Supplementary Fig. 2 and Supplementary Data 1). The yellow module, enriched for GO-

term functions related to localization and carboxylic acid synthesis, was upregulated in physogastric queens, QT3 and QT4 (yellow module, 537 genes; Supplementary Fig. 2 and Supplementary Data 1). The lightcyan module (3455 genes; Supplementary Fig. 2 and Supplementary Data 1) was the largest among the modules and enriched for functions related to transcription and general protein synthesis. It may contain the

co-expressed genes affecting long-lived reproductives of both sexes QT4 and KT4, though the latter only as a trend (KT4 0.39 *P* value = 0.05; Fig. 2 and Supplementary Fig. 2).

**Long-lived reproductives share expression patterns indicative of changed IIS signaling and prevention of several aging processes.** A comparison of transcriptomic profiles between fat bodies of long-lived reproductives (KT4 and QT4) and short-lived FW allowed us to look at genes potentially underlying their long lifespan in natural conditions, irrespective of reproductive efforts distinguishing kings from queens (Supplementary Fig. 3). A total of 1454 genes were upregulated in fat bodies of KT4 and QT4 in comparison to FW, including 350 genes that differed between KT4 and QT4. In addition, 2208 genes were downregulated in KT4 and QT4 relative to FW, 468 of which were significantly different between KT4 and QT4. By analyzing these differentially expressed genes (DEGs), we found evidence indicating that reproductives avoid several aging processes. For instance, the expression of many genes important for genome stability, including genes involved in DNA damage response and telomere maintenance, was upregulated in KT4 and QT4 (Fig. 3 and Supplementary Tables 2 and 3). In addition, several genes coding for the oxidative phosphorylation system (OXPHOS) and the mitochondrial ribosomal proteins, mitochondrial transport, and mitochondrial fission were upregulated in QT4 and KT4 (Fig. 3 and Supplementary Tables 2 and 3). This suggests that mitochondrial function was maintained in the fat bodies of long-lived reproductives with likely beneficial effects on cell integrity and oxidative status. In agreement with this, we observed a downregulation of the expression of several antioxidant genes in QT4 and KT4 relative to FW (Fig. 3 and Supplementary Tables 2 and 3). These results were accompanied by the upregulation of gene expression involved in several processes related to protein and macromolecule synthesis (Supplementary Fig. 3).

We observed significantly upregulated expression of upstream components of IIS signaling in the fat bodies of KT4 and QT4 relative to FW. This included an 800-fold increase in gene expression of an insulin-like peptide which we called *Ilp9* (Figs. 3 and 4), the catalytic subunit phosphatidylinositol 3-kinase *pi3k59F*, as well as two insulin targets, *eIF6* and the transcription factor *crc*[38] (Fig. 3 and Supplementary Tables 2 and 3). In contrast, the expression of several actors involved in the TORC1 signaling pathway (*tor*, *raptor* and its substrate, *S6K*) were downregulated in QT4 and KT4 relative to FW, as well as the *mdy* lipid metabolism gene (Fig. 3 and Supplementary Tables 2 and 3). *Mdy* encodes a diacylglycerol acetyltransferase which catalyzes the final step of triglyceride (TG) synthesis from diglycerides (DG)[39], and its downregulation suggests a decrease of fat storage in reproductives, to be confirmed by lipidomics below.

**Expression specific to highly fecund queens suggests further adaptations in IIS signaling and a metabolism geared toward oogenesis.** To better understand how highly fecund queens from mature colonies defy the reproduction/lifespan trade-off, we focused on gene expression patterns specific to QT4. For this, we concentrated on genes, for which gene expression significantly differed in QT4 from both FW and KT4, thus allowing insights into the allocation of resources towards oogenesis. Again, we found that several genes involved in the IIS pathway were differentially regulated in QT4 relative to FW and KT4. For instance, the two effectors of the IIS pathway discussed in the previous section, *pi3k59F* and *crc*, were not only upregulated in both reproductives but more strongly so in QT4 than in KT4 (Supplementary Table 3), suggesting an additional relevance of their

expression for female fertility. In fact, two further *pi3k*-genes showed QT4-specific expression. While the regulatory *pi3k21B* subunit was upregulated in the fat body of QT4 compared to FW, the catalytic *pi3k92E* was downregulated in QT4 relative to FW (Supplementary Table 3). The insulin receptor *InR3* and the kinase *pdk1* were upregulated in QT4 relative to FW but downregulated in KT4 (Supplementary Table 4).

We also found major gene expression differences in carbohydrate and lipid metabolism pathways, which are known to be activated by the IIS pathway[2]. Several genes, involved in the glycogenesis and trehalose energetic storage pathways, were upregulated in FW (Supplementary Tables 2–4). Trehalose-6-phosphate synthase was downregulated in QT4 relative to FW and KT4 (Supplementary Table 4). In QT4 queens, on the other hand, several genes which are key players in carbohydrate catabolism were upregulated relative to KT4 and FW, particularly genes encoding enzymes involved in glycolysis (e.g., *cg6650* also known as *adpgk*, *pfk*, and *pkm*; Supplementary Table 4), as well as the hexosamine biosynthetic pathway (HBP) and pentose phosphate pathway (PPP) (Supplementary Tables 3 and 4). Queens also showed upregulation relative to FW and KT4 of genes promoting fatty acid (FA) synthesis from carbohydrates (*acc* and *fasn1*; Supplementary Table 4) and FA activation, esterification, and elongation (Supplementary Table 4). Heightened DG transport in QT4 is suggested by an upregulation relative to FW and KT4 of expression of lipoprotein genes essential for oogeneses, such as the female-specific vitellogenin (*vg*) and the diacylglycerol-carrying lipoprotein (*hdlbp*) (Supplementary Tables 3 and 4).

**Transcriptomic analysis of queen maturation stages suggests different timings of activation of oogenesis and lifespan-preserving mechanisms.** Concomitantly with the growth of colonies and coinciding with their development of physogastry, the imaginal fat body of virgin queens (QT0) with its canonical fat storage function becomes replaced by a royal fat body (QT4), which is highly oriented towards specific protein synthesis and secretion[40]. A comparative analysis of different termite species recently suggested that endopolyploidy (or nuclear genome replication without cell division) permits the rate of vitellogenin synthesis to increase[41]. Developmental changes in oogenesis might rely on endopolyploidy and we therefore determine the percentages of the nuclei count at each ploidy level (2C, 4C, and 8C) in the fat bodies of adult queens in five stages. To investigate when molecular mechanisms affecting lifespan and oogenesis are activated, we compared gene expression between adult queens in these five stages (Fig. 1). Different regions of the gene co-expression network were upregulated in each stage (Supplementary Fig. 4). Next, we focused on comparing expression differences between adjacent stages.

First, levels of expression of *Ilp9*, *InR3*, and *vg* genes were not different between the physogastric queen stages QT3 and QT4 (Supplementary Data 2 and Fig. 5). However, *crc* gene expression (Supplementary Data 2) and genes involved in glycolysis and OXPHOS, mitochondrial membrane transport proteins, and mitochondrial ribosomes were all strongly upregulated in QT4 relative to QT3 (Supplementary Data 2). We can conclude from this that among mature physogastric queen stages, substantial upregulation of important lifespan-preserving processes occurs with age but not of processes linked to oogenesis. In particular, mitochondrial functions seem to be upregulated substantially with age.

Due to a lack of workers which feed all colony members[40], queens initially live several weeks after mating without food. In the imaginal fat body of virgin queens (QT0), GO analysis

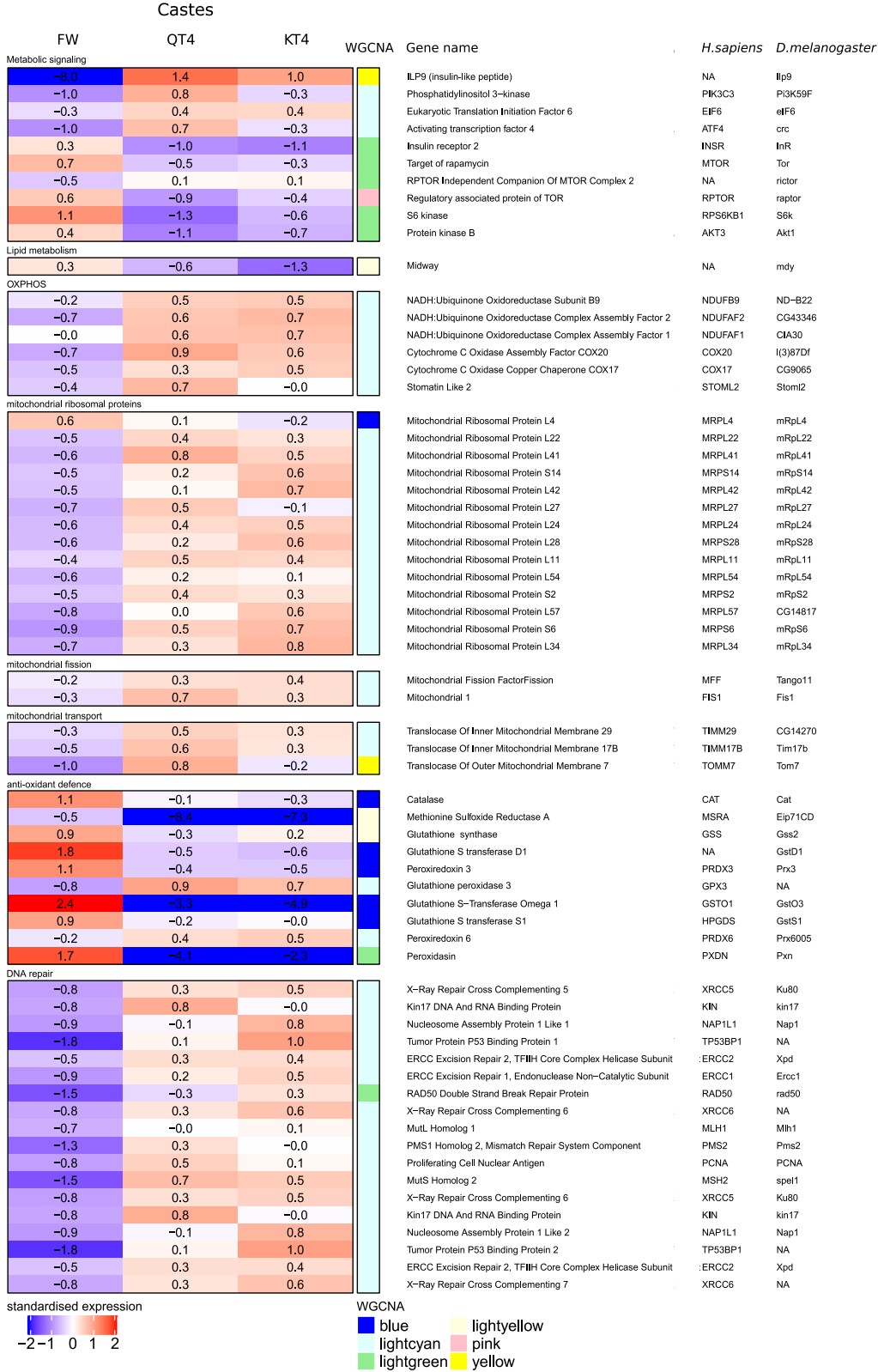

**Fig. 3 Caste-related changes in gene expression.** Heatmap representing standardized gene expression (blue = low; red = high) in fat bodies of FW, QT4, and KT4. Annotations to the right of the heatmap include the WGCN module (Supplementary Fig. 2), gene names and gene acronyms in *Drosophila melanogaster* (*D. melanogaster*) and *Homo sapiens* (*H. sapiens*). The map is restricted to the expression of genes involved in metabolic signaling, lipid metabolism, mitochondrial oxidative phosphorylation system (OXPHOS), mitochondrial ribosome proteins, mitochondrial fission, mitochondrial transport, antioxidant defense, and DNA repair. Expression of all genes differs significantly between FW versus QT4 and between FW versus KT4. The number of replicates per group is provided in Supplementary Table 1.

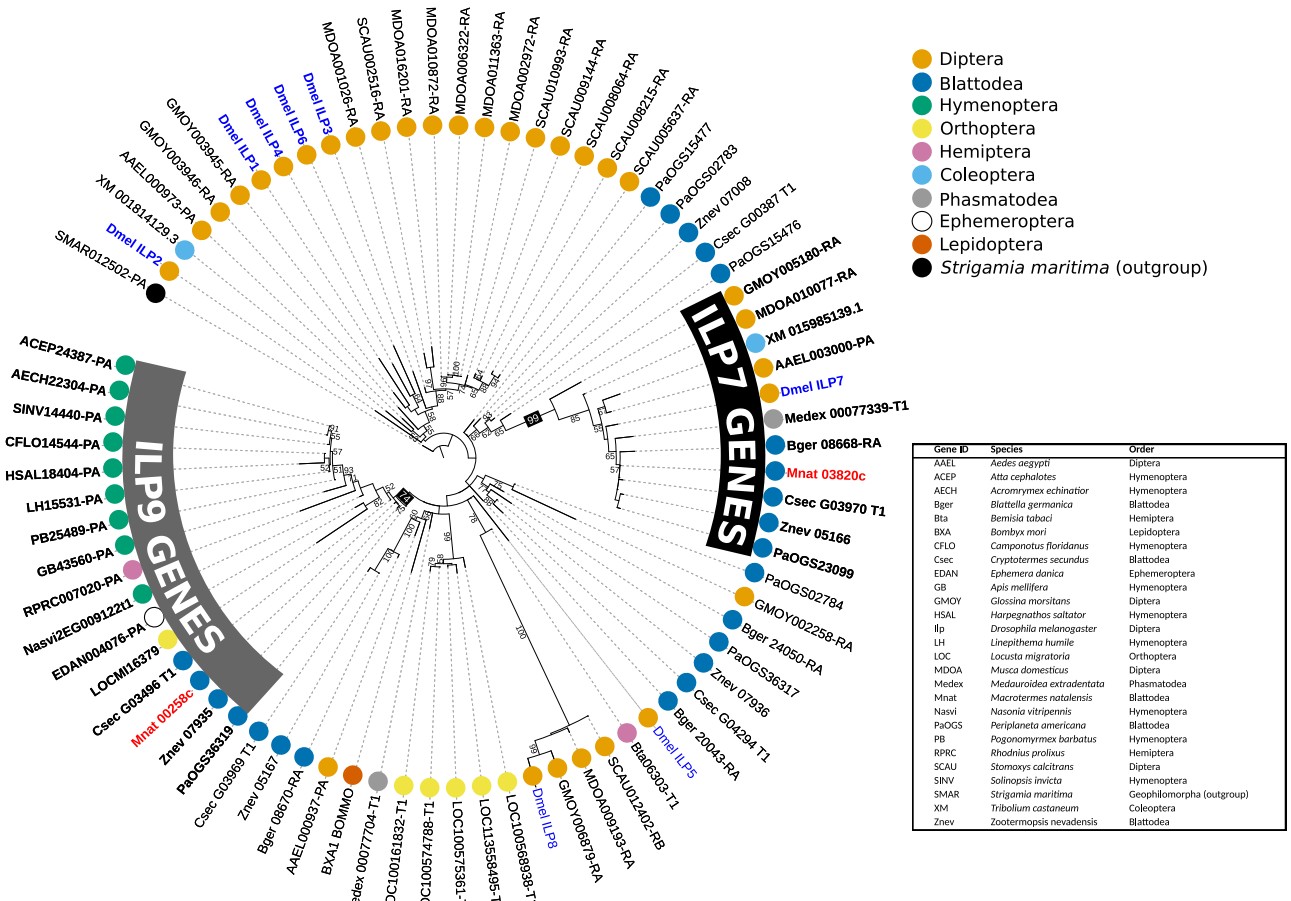

**Fig. 4 Phylogenetic tree of ILP genes in several insect species.** The two ILPs found in the *M. natalensis* genome (*Ilp7* = Mnat03820c and *Ilp9* = Mnat00258c) are highlighted in red and the eight Drosophila genes (*DILPs*) are highlighted in blue. All other species can be identified by their GeneID displayed in the Table. The tree was constructed with iqtree2 (Q.pfam+R4 model and 10,000 bootstraps) and visualized with iTOL (v. 6). Branch labels are bootstraps, the tree is rooted at the *Strigamia maritima* gene.

revealed that several signal transduction pathways involved in fat body cell transformation and apoptosis were upregulated relative to queens at the end of this starvation period (stage QT1; Supplementary Fig. 4 and Supplementary Data 2). Furthermore, within the IIS pathway, the expression of *InR2*, *InR3*, *pi3k21B*, and *Akt1* were downregulated in QT1 relative to QT0 (Supplementary Data 2). At QT1, we observed an enrichment of GO terms related to an increase of catabolic processes such as proteolysis and autophagy and an upregulation of genes involved in pathways linked to an increased use of fatty acid reserves by β-oxidation (Fig. 5, Supplementary Fig. 4, and Supplementary Data 2). We observed further responses which can be explained as stress responses to starvation and which involve mechanisms known to affect aging. AMP-activated protein kinase α (*Ampkα*), which is known to be activated under conditions of low energy and initiates both the degradation of damaged mitochondria and mitochondrial biogenesis[42], was significantly upregulated in QT1. We observed an upregulation of two genes (*drp1* and *fis1*) known to facilitate mitochondrial degradation by autophagy and two genes involved in mitophagy (*cg5059* and *pink1*), a specific degradation of mitochondria via autophagy (Supplementary Data 2). Moreover, we observed an upregulation of the expression of genes involved in the OXPHOS system, mitochondrial ribosome subunits, and mitochondrial transport proteins, suggesting an increase in mitochondria biogenesis (Supplementary Data 2). Finally, we observed an increased expression of antioxidant genes (Supplementary Data 2) which are known to

be upregulated by AMPK pathway in situations of oxidative stress[43].

Given the increase in the number of workers per colony and of fungus development ("Methods" and Supplementary Fig. 5b, c), we believe that QT2 are fed by workers. At QT2, antioxidant-defense processes, as well as *Ampkα* gene expression were downregulated (Supplementary Data 2). At this stage, GO terms showed that several upregulated processes were linked to protein metabolism (Supplementary Fig. 4). *Ilp9*, *pi3k21B* gene expression, and de novo lipogenic genes were upregulated in QT2 compared to QT1, suggesting a first activation of the IIS pathway components in the fat body of QT2 (Supplementary Data 2 and Fig. 5). At the same time, we observed a downregulation of *mdy* gene expression, even relative to the physogastric queen stages (Fig. 5). In addition, several genes involved in the cell cycle, including *mcm2, 5, 6, 7*, and *10*, several cyclins and cyclin-dependent kinases, such as *cycE/cdk2* and *orc1*, and the proto-oncogene *myc* were upregulated (Supplementary Data 2). Simultaneously with the increase of *vg* expression in QT2 fat bodies, we observed an increase of polyploidy levels. The proportion of 4C nuclei now exceeded 70% (Supplementary Fig. 6) in comparison to 35% in QT0 and QT1. Taken together, these data suggest that QT2 is a transitory stage where the final phase of fat body maturation is prepared or initiated in many processes. Our PCA results support these effects (Supplementary Fig. 1b).

In QT3, the abdomen has become enlarged. We found that over 90% of all cells in the fat bodies of QT3 and QT4 have 4C

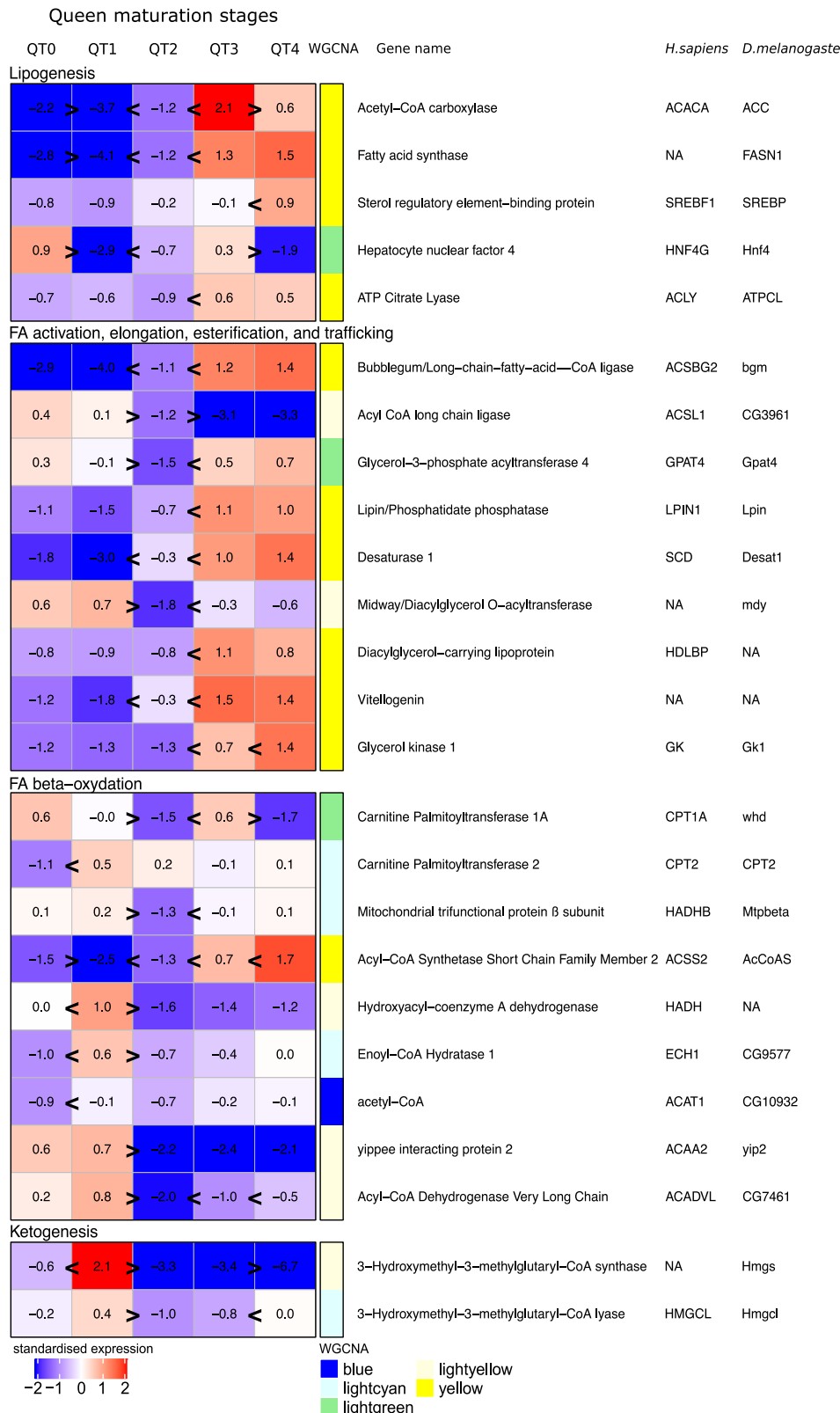

**Fig. 5 Heatmap with the expression of genes in fat bodies during adult queen maturation involved in lipid metabolism.** Heatmap representing standardized gene expression (blue = low; red = high) at each of the five queen stages (QT0–QT4). Annotations to the right of the heatmap include the WGCN module (Supplementary Fig. 2), gene name, and gene acronyms in *Drosophila melanogaster* (*D. melanogaster*) and *Homo sapiens* (*H. sapiens*). The number of replicates per group is provided in Supplementary Table 1. Greater than or less than symbols (>/<) represent significant differences in expression.

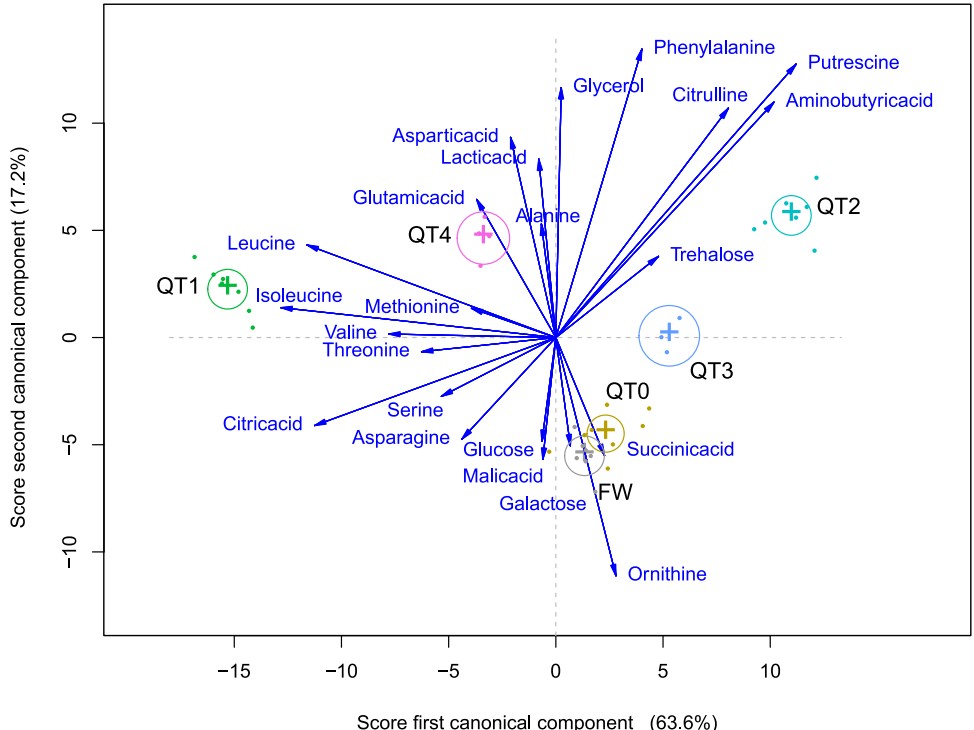

**Fig. 6 Canonical discriminant analysis of concentrations of metabolites.** Three canonical functions discriminate ages and castes significantly, of which we show scores for the first two. Average scores for each caste and age are shown, plus canonical structure coefficients of each metabolite as vectors from the origin. These are proportional in length to the magnitudes of the correlations of each metabolite with the scores of the discriminant functions and show how information from each metabolite aids in discriminating castes and ages. The numbers of replicates per group are provided in Supplementary Table 1.

nuclei (Supplementary Fig. 6). During this period, expression of *Ilp9* and *InR3* genes was upregulated in comparison to QT2 whereas *InR2* gene expression decreased (Supplementary Data 2). Genes involved in glycolysis, HBP, and PPP (Supplementary Data 2) were upregulated. Similarly, the expression of genes involved in de novo lipogenesis, as well as FA activation, elongation, esterification, and transport were upregulated (Fig. 5). These expression patterns did not change further in QT4.

Overall, our data suggest that the QT2 stage acts as a transitional period where *Ilp9* and specific downstream IIS pathway components become upregulated. This is associated with a strong downregulation of the *mdy* gene involved in triglyceride storage. The expression patterns corresponding to increased oogenesis are observed when physogastry is present. Several important lifespan affecting processes seem to be upregulated in QT4, long after maturation.

**Major changes in metabolites and lipid composition highlight a lack of triglycerides and an abundance of glucose in long-lived queens.** We performed metabolomic analysis on hemolymph samples of workers and queens in different stages (FW and QT0–QT4, sampling in Supplementary Table 1) to confirm and extend the results on expression patterns of genes involved in carbohydrate and lipid metabolism. Metabolomes differed between FW and QT4, and between each pair of successive queen maturation stages (pairwise perMANOVA, each *P* value <0.05; Fig. 6). FW and QT0 seemed little different overall, while starved QT1 were well separated from the QT2 individuals in transition toward massive reproduction. Physogastric QT3 was globally intermediate between QT0, QT2, and QT4 (Fig. 6). We compared individual metabolite concentrations between FW and QT4 and between successive queen maturation stages (Supplementary Fig. 7 and Supplementary Table 5). We highlight the importance

of simple sugars. Glucose and galactose are the metabolites present in the largest concentrations in QT4. This high concentration of glucose[44] occurred similarly in FW but not in any of the younger queen stages. Confirming our gene expression findings, the concentration of trehalose was lower in QT4 than in FW, but surprisingly also lower in QT4 than in any other queen stage (Supplementary Fig. 7 and Supplementary Table 5). Alanine, glycerol, aspartic acid, phenylalanine, and glutamic acid occurred in larger concentrations in QT4 than FW (Supplementary Fig. 7 and Supplementary Table 5), corresponding with ongoing increased protein turnover and lipid synthesis. Relative to QT0, starved queens (QT1) had increased levels of leucine, threonine, isoleucine, and citric acid, which decreased again in QT2 (Supplementary Fig. 7 and Supplementary Table 5). This is indicative of proteolysis during starvation.

We performed a lipidomic analysis in hemolymph of FW and QT4 and in fat bodies of QT0, QT2, and QT4 (Supplementary Table 1). The lipidomic analysis on hemolymph revealed 81 esterified lipid species of which 34 differed in quantity and composition between FW and QT4 (Fig. 7 and Supplementary Table 6). Strikingly, DG were significantly elevated in QT4 relative to FW while several TG were significantly diminished (Fig. 7 and Supplementary Table 6). This decrease of TG concentrations in QT4 hemolymph as well as the upregulation of genes involved in the IIS pathway and de novo lipogenesis in fat bodies, plus the decrease of *mdy* gene expression observed in reproductives relative to FW, suggest a decrease in fat body TG storage in QT4. Thirteen TG concentrations were in fact significantly lower in QT4 fat bodies relative to QT0 (Fig. 8a and Supplementary Table 7).

In addition, we determined fatty acid composition in fat bodies of queens to assess the levels of oxidative cell damage. We investigated changes in proportions of polyunsaturated fatty acids (PUFA), monounsaturated fatty acids (MUFA) and saturated

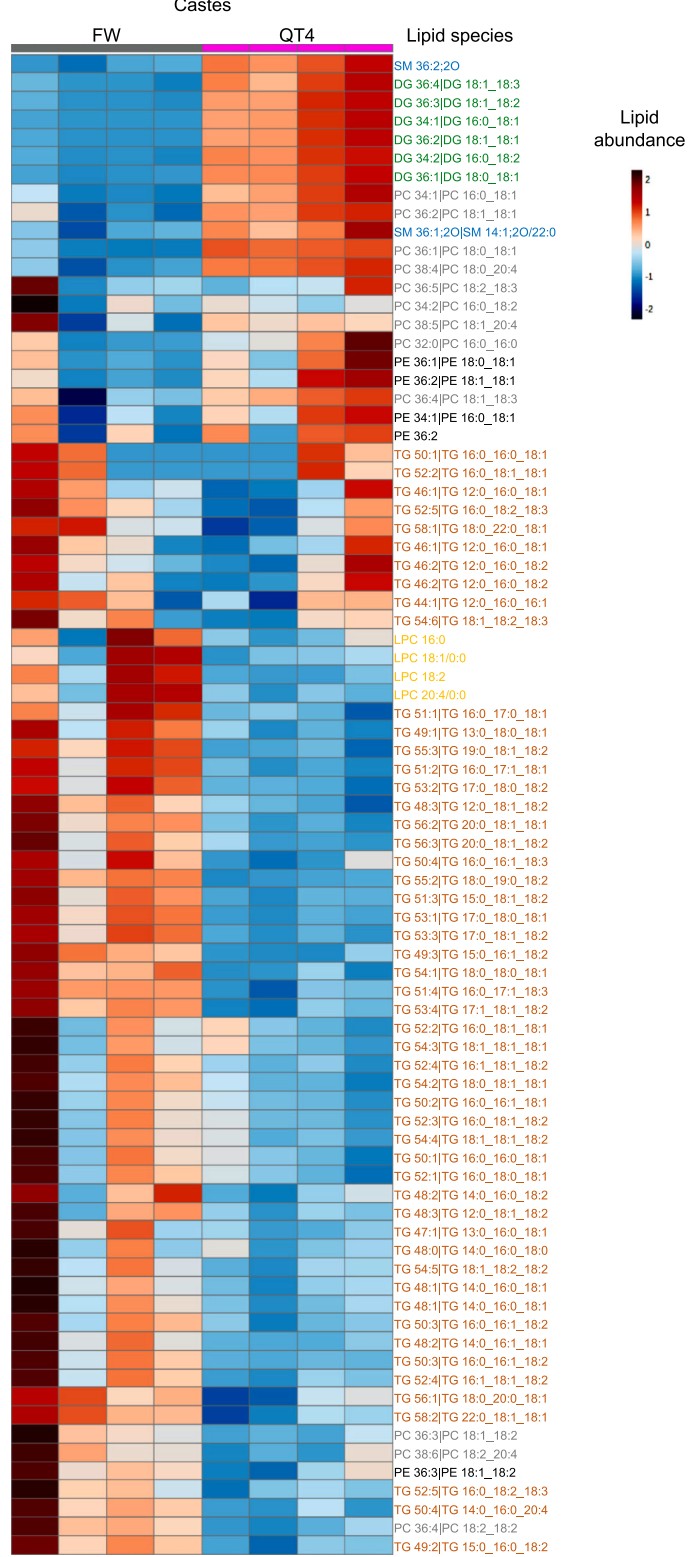

**Fig. 7 Comparison of lipid profiles between FW and QT4 in hemolymph.** Hierarchical clustering heatmap analysis of triglycerides (TG, orange), diglycerides (DG, green), phosphatidylethanolamine (PE, black), phosphatidylcholine (PC, gray), lysophosphatidylcholine (LPC, yellow), sphingomyelin (SM, blue) lipids in hemolymph of FW and QT4 performed in MetaboAnalyst 4.0. Individual lipids are shown in rows and samples displayed in columns, according to cluster analysis (Euclidian distance was used and Ward's clustering algorithm). The color gradient, ranging from dark blue through white to dark red, represents low, middle, and high abundance of a lipid. Numbers of replicates per group are provided in Supplementary Table 1.

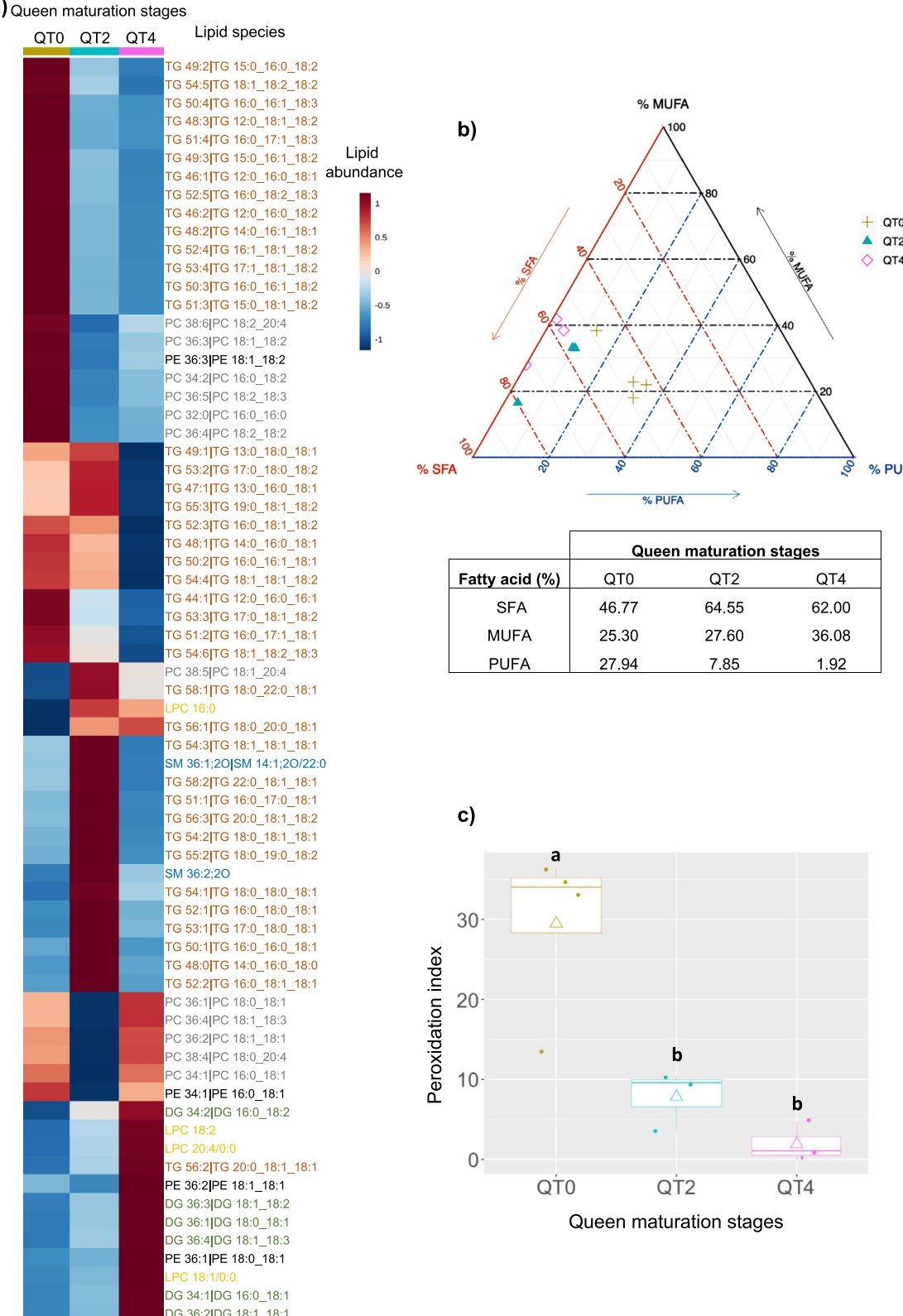

| | Queen maturation stages | | |
|---|---|---|---|
| **Fatty acid (%)** | QT0 | QT2 | QT4 |
| SFA | 46.77 | 64.55 | 62.00 |
| MUFA | 25.30 | 27.60 | 36.08 |
| PUFA | 27.94 | 7.85 | 1.92 |

fatty acids (SFA) in fat bodies of QT0, QT2, and QT4. PUFA/MUFA/SFA proportions in fat bodies of QT2 and QT4 were significantly different from QT0 (PERMANOVA, *P* value <0.05; Fig. 8b). This difference seemed to be related to a larger proportion of PUFA (highly oxidizable) in QT0 compared to QT2 and QT4 (Fig. 8b). We calculated the relative peroxidability of membranes (peroxidation index, PI), which decreased drastically from 29.5 at QT0, 7.9 at QT2 to 1.9 at QT4 (Fig. 8c). This can explain the scope for the downregulation of antioxidant genes in QT4.

## Discussion

Our results reveal several mechanisms by which long-lived termite reproductives defy aging. They support the joint occurrence

**Fig. 8 Changes in lipid profiles in fat bodies during adult queen maturation. a** Hierarchical clustering heatmap analysis of triglycerides (TG, orange), diglycerides (DG, green), phosphatidylethanolamine (PE, black), phosphatidylcholine (PC, gray), lysophosphatidylcholine (LPC, yellow), sphingomyelin (SM, blue) lipids in fat body of different stage of the queen (QT0, QT2, and QT4) performed in MetaboAnalyst 4.0. Individual lipids are shown per row and mean of lipid amount of each stage displayed in columns, according to cluster analysis (Euclidean distance and Ward's algorithm). The color gradient, ranging from dark blue through white to dark red, represents low, middle, and high abundance of a given lipid. **b** Percentages (%) of saturated fatty acids (SFA), monounsaturated fatty acids (MUFA), and polyunsaturated fatty acids (PUFA) of total FA in fat bodies of different queen maturation stages (QT0, QT2, QT4). Ternary graph showing the percentages of SFA (orange), MUFA (black), and PUFA (blue). The table below shows the averages of MUFA, SFA, and PUFA for each stage. Permutational MANOVA demonstrated that SFA/MUFA/PUFA proportions in fat bodies of QT2 and QT4 were significantly different relative to QT0 (permutational MANOVA, *P* value = 0.012). **c** Box plot illustrating the peroxidation index of fat bodies of different queen maturation stages (QT0, QT2, and QT4). A box consists of upper and lower hinges and a center line corresponding to the 25th percentile, the 75th percentile, and the median, respectively. Rhombuses represent the averages. Different letters indicate significantly different values according to a Kruskal–Wallis test followed by pairwise Dunn tests (*P* values <0.05). The number of replicates per group is provided in Supplementary Table 1.

of the avoidance of DNA damage accumulation, the maintenance of mitochondrial functioning and turnover, accompanied by a reduction of antioxidant defenses. Termite queens sustain high fertility throughout their long life without apparent costs of their long-term massive reproductive output. We observed a remodeling of signaling pathways across queen maturation stages which could help to prevent hyperfunction[45] despite sustained insulin signaling. The composition of lipids in long-lived queens is less sensitive to oxidative damage and the observed scarcity of stored fat (TGs) observed relative to DGs seems to optimize both sustained fecundity and healthspan. All this seems possible while termite queens are fed mostly on simple sugars by female workers.

**Long lifespan on a carbohydrate diet**. Overall, all results confirm that long-lived queens receive a highly energetic food enriched in simple sugars. In many organisms from *Drosophila* to humans, a prolonged carbohydrate-rich diet is associated with chronic metabolic diseases reducing lifespan[29,46]. Increased insulin secretion caused by such a diet leads to the accumulation of TG, and disrupted IIS signaling over time leads to the development of insulin resistance[2,46–48]. The intermediary metabolism in QT4 queens is centered on a high use of carbohydrates with an increase of glycolysis, de novo lipogenesis, and the OXPHOS system for energy generation and the synthesis of specific lipids and proteins. Poulsen et al.[49] have additionally shown that the diversity of decomposition enzymes encoded by the queen's microbiota is low and geared towards hydrolyzing simple sugars. The unexpected low trehalose concentrations in long-lived queens, coupled with their high glucose levels also occurring in female workers suggest that glucose is an important component of what female workers feed to physogastric queens in colonies with a well-developed fungus, without much trehalose synthesis or prompt trehalose hydrolysis. In mammals and even in *C. elegans*, trehalose counteracts disruptions of protein homeostasis by oxidative stress, temperature variation and dessication[50–52] and thus leads to increasing lifespan. The absence of trehalose in long-lived queens is possibly facilitated by the protected lifestyle of reproductives in the royal cell, or by compensatory mechanisms such as the maintenance of a healthy mitochondria population producing low amounts of ROS[53–56]. We found expression patterns in trehalose pathways that were specific to physogastric queens, suggesting that the patterns of hemolymph sugars in kings might be less extreme.

**No fat storage in long-lived queens**. Remarkably, we observed a transcriptional downregulation of genes coding for different mechanisms of energy storage (triglycerides, glycogen, and trehalose) in long-lived queens. Using resources immediately to sustain fecundity is detrimental in most iteroparous life-history

strategies. Most individuals need to maintain reserves themselves, for example, in order to buffer periods with low foraging success (starvation)[30,44]. In social insects, this task can be delegated to workers assuring a constant food supply.

Coinciding with upregulated de novo lipogenesis and upregulation of the *vg* gene from QT2 queens onwards, expression of the *mdy* gene coding a diacylglycerol acyltransferase was downregulated. This can explain the decrease of TG concentrations we observed in long-lived queens and suggests very limited storage of TG in physogastric queens in favor of immediate utilization. Also, we observed a trend for DG to increase, which are known to be preferably used for lipid transport[57] (not free fatty acid as in vertebrates). Han & Bordereau[40] observed almost 40 years ago low levels of lipid droplets in the fat body of long-lived *Macrotermes* termite queens. A decrease of *mdy* gene expression was also observed in the fat body of mature kings, suggesting that a low level of stored TG could be generally beneficial for long lifespan in reproductives and is not just explaining the absence of a cost of reproduction in queens. Our results make the gene network involving *mdy* and fat concentrations in reproductives of both sexes targets for research linked to the accumulation of excess fat.

**Surprising upregulation of IIS pathway components in reproductives**. A comparative analysis of termites, ants, and bees suggested that downstream components of the IIS/TOR signaling pathways play a consistent role in determining lifespan in eusocial insects[16]. We found changes in the IIS pathway in the fat bodies of reproductives. In *Drosophila*, seven out of eight insulin-like peptides (DILPs) are mainly produced in the brain. The exception, DILP6, is produced in the fat body and its gene expression is upregulated during starvation[58,59]. In *M. natalensis*, we identified two genes from separate loci, one coding an ortholog to DILP7 (not expressed in *M. natalensis* fat bodies) and a further paralog found across holo- and hemimetabolous insects, without a clear ortholog in *Drosophila melanogaster*, which we called *Ilp9*. During queen maturation, we observed potential increases in insulin suggested by greatly upregulated expression of *Ilp9* in the transitional QT2 stage and when physogastry becomes established. It is ~800-fold more expressed in fat bodies of long-lived reproductives relative to female workers. The increase of *Ilp9* gene expression in long-lived reproductives was not associated with an activation of TORC1, although the expression of genes involved in protein synthesis was upregulated in their fat bodies. We observed an upregulation of *elF6* gene expression which is involved in insulin-stimulated translation, most notably by controlling adipogenic transcription factors like *crc* (also known as *ATF4*), a member of the mTOR-independent pathway[38,60]. We propose that *Ilp9* activates the eIF6-*crc* gene program in the fat bodies of mature queens and kings. This in turn increases the synthesis of proteins that are involved in lipid

synthesis and essential for fecundity. This occurs despite down-regulation of TORC1, elsewhere described as the main pathway for protein synthesis[61] and also found downregulated in whole bodies of two lower termite species[24,25].

**Maintenance of mitochondrial turnover with low ROS production in long-lived reproductives.** Several studies demonstrated that during calorie restriction, mitochondrial stress induced by a transitory increase of ROS leads to a cellular adaptive response named mitohormesis which, in the long term, allows for more effective stress resistance and lifespan extension[62–64]. During the starvation period at the QT1 stage, the absence of nutrition was associated with an upregulation of genes coding for mitochondria biogenesis (OXPHOS system, mitochondrial ribosome subunits, and transport) and mitochondrial fission (a mechanism involved in mitochondrial degradation by autophagy). This allows an increase in stress resistance by increasing both the degradation of damaged mitochondria and the synthesis of new ones[62]. We also observed an upregulation of the expression of genes coding for antioxidant enzymes, which may suggest a mitohormetic response to an increase of ROS production. Interestingly, we observed that genes coding mitochondrial synthesis and fission were again upregulated in QT4 and KT4 reproductives, suggesting a maintenance of mitochondrial turnover while genes coding for antioxidant enzymes were downregulated. Similar downregulation was observed in honeybee and ant queens[65,66]. Our findings suggest that the mitochondria of long-lived reproductives remain efficient and produce a comparatively low amount of ROS, thus preventing oxidative damage and provoking at most a weak mitohormetic response.

**Lower sensitivity to oxidative damage in long-lived queens.** Several studies in mammals, birds, and invertebrates[67,68] have demonstrated that PI values correlate negatively with longevity. We highlighted significant decreases in the fraction of poly-unsaturated fatty acids (PUFA) in QT2 and again in QT4 queens, leading to a strong, progressive decline of the peroxidation index (PI). PUFA are a thousand times more likely to oxidize than MUFA[67]. Their oxidation can set off an oxidative cascade with the formation of radicals capable of damaging surrounding macromolecules and tissues[67]. Therefore, the decreases in PUFA fractions can lead to a reduction in oxidative damage and increase lifespan of queens while genes coding for antioxidant enzymes are downregulated. We therefore predict that PI values decrease similarly in kings, which could be confirmed by future studies on different maturity stages of kings.

In conclusion, our study indicates that developmental plasticity in this natural system allows it to overcome several well-known hallmarks of aging[3], including mitochondrial dysfunction, genomic instability and deregulated nutrient sensing. Our results highlight the importance of an insulin-like peptide (ILP9), which is upregulated in the fat bodies of long-lived reproductives and likely changes downstream signaling, leading to the gene expression profiles and metabolism which extend lifespan. These findings have implications for aging research, supporting the notion that aging can only be arrested when all factors which increase mortality over time are addressed.

## Methods
**Sampling.** *Macrotermes natalensis* lives in large colonies in Southern Africa where it builds massive mounds[69]. Field colonies opened to collect animals had been followed for over 20 years by Jannette Mitchell in an experimental field of the University of Pretoria (coordinates in Supplementary Table 8)[70]. Old minor adult workers (FW), virgin queens (QT0), 20-year old queens (QT4), and kings (KT4) were sampled from at least 20-year-old colonies. Less than an hour was taken to reach the royal cells containing QT4 and KT4. QT4 and KT4 also showed limited variability in weight and length between colonies. Concerning insects, no ethical

approval was required. The study was conducted according to the Nagoya protocol. Samples (tissues and hemolymph) were exported at −80 °C from Pretoria-South Africa to Bondy-France (permit 93010001).

**Establishment and maintenance of incipient termite colonies.** When natural colonies are mature (5–7 years after establishment) and in appropriate environmental conditions, winged male and female imagoes leave the mound during spring and disperse in synchronous swarms[70]. Imagoes were collected in Pretoria (South Africa) in 2016 and 2018 during the spring swarming flights (coordinates in Supplementary Table 8). Mounds were covered with nets to retrieve imagoes. These were placed in large boxes preserving humidity and immediately transferred to the laboratory. In the field, males locate female imagoes[71] and paired couples perform dealation and establish new colonies as queen and king[70]. Imagoes collected were sexed by visual observation of their abdominal sternites. Weight and length were recorded and wings were manually removed. The establishment of laboratory incipient colonies occurred for both field trips, following a protocol adapted from Lepage[36] and Han & Bordereau[20] (Fig. 1 and Supplementary Fig. 8). We established 1600 incipient colonies. Each paired couple was introduced in a closed plastic box (6 × 5 × 4.5 cm) filled with sieved soil collected near the mounds. The incipient colonies were kept in a breeding room with controlled conditions: 28 °C, 85% relative humidity, and 12:12 photoperiod. Water was used to keep the soil slightly moistened. The development of the colonies was visually monitored. In the field, workers explore the environment a few months after colony establishment to collect spores and to inoculate the fungus comb they build in the nest[72]. Workers feed all colony members through trophallaxis (transfer of food from the mouth to mouth), after a complex digestion of lignocellulose by the fungus and intestinal microbiota[49]. At 3 months in the laboratory colonies, when workers started to explore, small pieces of dry wild oats were supplied on the surface of the soil, and wood was additionally supplied after 4.5 months. A *Termitomyces* sp. fungus comb with nodules was collected from one mature field colony and a small part of this comb was introduced in each box. After 3.5 months, mortality was 56% for the 2016 incipient colonies (±15% across field colonies of origin) and 30 ± 7% for the 2018 incipient colonies. When the termite populations outgrew their boxes, they were opened on one side and placed inside bigger ones (18 × 12 × 7.5 cm after three months, 36 × 24 × 14 cm after 14 months, and 1000 × 70 × 40 cm after 21 months) filled with sieved moistened soil. Colonies were checked every 2 days to supply water and food if needed and to remove moldy food. Queens were sampled after 3 months (QT1), 9 months (QT2) and after 31 months (QT3). During the first years of a colony's life, larvae emerge which become sterile workers or soldiers[35]. At QT1, the number of minor and major workers, soldiers, presoldiers, and larvae were counted in colonies where queens were sampled. At QT2, these were also counted and the fungus was weighed.

**Species identity.** Total DNA was isolated from the head and the legs of one imago of each of the nine colonies. PCR was performed using the cytochrome oxidase I gene primers: LCO 5'-GGT CAA CAA ATC ATA AAG ATA TTG G-3' and HCO 5'-TAA ACT TCA GGG TGA CCA AAA AAT CA-3'[73] and the 650-bp amplified fragment was sequenced and analyzed using the Barcode of Life Database identification system (www.barcodinglife.org). Species identity of each colony was confirmed to be *M. natalensis*[49].

**Hemolymph and fat body collection.** Fat bodies were collected from cold-anesthetized individuals (FW, QT0, QT1, QT2, QT3, QT4, and KT4). Hemolymph was collected on FW, QT0, QT1, QT2, QT3, and QT4 (Supplementary Table 1) under a binocular microscope with tapered glass Pasteur pipettes inserted in the membranous part just behind the head. The mean volume of hemolymph collected per individual was 0.5 μL for FW, 1.5 μL for QT0, QT1, QT2, 50 μL for QT3, and 1 mL for QT4. Hemolymph samples were collected in cryotubes, quickly frozen in liquid nitrogen and kept at −80 °C until use. Subsequently, termites were killed by decapitation and their abdominal fat bodies were collected. For RNA and DNA extraction, the fat body was stored in a tube containing RNAlater buffer (Invitrogen) and kept at −80 °C until use. For lipid and metabolite analyses, nitrogen-frozen fat bodies were crushed in a tube which was immediately frozen in liquid nitrogen and kept at −80 °C until use. For ploidy analyses, the fat body was collected from one individual, stored in a tube containing 200 μL of Cycletest PLUS DNA Reagent Kit buffer (Becton Dickinson), and kept at −80 °C until use.

**RNA profiling.** In the 2016 and 2018 cohorts, total RNA was isolated from fat bodies of FW, QT0, QT1, QT2, QT3, QT4, and KT4 using miRNeasy Micro kit (Qiagen) and RNAse-free DNAse according to the manufacturer's instructions (Qiagen). The number of replicates per group is provided in Supplementary Table 1. RNA-Seq library preparations were carried out from 500 ng total RNA using the TruSeq Stranded mRNA kit (Illumina, San Diego, CA, USA) which allows mRNA strand orientation (sequence reads occur in the same orientation as anti-sense RNA). Briefly, poly(A)+ RNA was selected with oligo(dT) beads, chemically fragmented and converted into single-stranded cDNA using random hexamer priming. Then, the second strand was generated to create double-stranded cDNA. cDNA were 3'-adenylated and Illumina adapters added. Ligation products were PCR-amplified. All libraries were subjected to size profile analysis conducted

by Agilent 2100 Bioanalyzer (Agilent Technologies, Santa Clara, CA, USA) and qPCR quantification (MxPro, Agilent Technologies, Santa Clara, CA, USA) using the KAPA Library Quantification Kit for Illumina Libraries (KapaBiosystems, Wilmington, MA, USA), then sequenced using 150-bp paired-end read chemistry on a HiSeq 4000 Illumina sequencer (Illumina, San Diego, CA, USA). An Illumina filter was applied to remove the least reliable data from the analysis. The raw data were filtered to remove any clusters with too much intensity corresponding to bases other than the called base. Adapters and primers were removed on the whole read and low-quality nucleotides were trimmed from both ends (when the quality value was lower than 20). Sequences between the second unknown nucleotide (N) and the end of the read were also removed.

*RNA-seq analyses*. The *M. natalensis* genome[49] was downloaded from the gigadb database (http://gigadb.org/dataset/100057; accessed March 2019). RNA-seq reads were mapped against the genome using *hisat2* (version 2.1.0[74]) at default settings. Gene expression levels were then generated by counting reads mapping to each gene of the *M. natalensis* genome (annotation version 2.3) using *htseq-count*[75]. Differential expression analyses were carried out in R (3.5.1) with the DESeq2 package[76], comparing between all pairs of castes and queen stages, as well as comparing each caste and queen stage against all others. Genes were considered significantly differentially expressed if the adjusted *P* value was less than 0.05. Principal component analyses (PCA) were also carried out within the DESeq2 package[76]. Counts were transformed using the *varianceStabilizingTransformation* function, and the PCA was calculated and plotted using the *plotPCA* function. This function carries out a PCA on the top 500 genes, based on variance. A weighted gene co-expression network (WGCN) was generated with these gene expression counts, using the R package WGCNA[37]. Normalized counts were extracted from the DESeq2 dataset with the *counts* function. These data were filtered for genes with zero variance or with missing values with the WGCNA function *good-SamplesGenes*. With the remaining 9631 genes, a signed WGCN was created using a soft power of 14, implementing the biweight midcorrelation calculation and setting the minimum module size to 30. Modules with a dissimilarity less than 0.5 were merged using the *mergeCloseModules* function. We related the expression profiles of the resulting nine modules to castes and queen stages by correlating (Pearson's *r*) the module eigengenes (first principal component of the expression matrix of each module) with a binary vector, containing 0 s and 1 s depending on the membership of each sample (FW, QT0, QT1, QT2, QT3, QT4, or KT4). A significant positive correlation signifies an overall upregulation while a negative correlation signifies a downregulation of expression within the module for a given caste or queen stage.

To visualize the WGCN, we first reduced the WGCN to include only the most highly connected nodes. We did this by retaining genes with a topological overlap of at least 0.2 with at least another gene, and by including the top 15 most connected genes within each module. This reduced WGCN (5823 genes) was exported to Cytoscape (version 3.8.0[77]) with the *exportNetworkToCytoscape* function in WGCNA (threshold 0.15). In Cytoscape, the network was rendered using the Edge-weighted Spring Embedded Layout and nodes were colored by module membership or expression fold change. GO-term enrichment analyses were carried out with topGO (version 2.34.0[78]), using the classic algorithm. Node size was set to 5, Fisher exact tests were applied, and we only kept GO terms that matched with two genes at least and with a *P* value <0.05. We established orthology of *M. natalensis* genes to *D. melanogaster* (v. 6.12) and *H. sapiens* (hg38) using the method of reciprocal best BLAST hit[79]. For this, the proteomes were blasted against each other using BLASTp (BLAST 2.7.1 + [80]) and an *e*-value threshold of 1e−5. Reciprocal best BLAST hits were extracted from the output files using a custom python script.

**Analysis of ILPs**. Two ILP genes (Mnat_00258 and Mnat_03820) were found in the *M. natalensis* proteome based on sequence similarity to ILPs in *D. melanogaster*, using BLASTp (BLAST 2.7.1 + [80]) with a *e*-value threshold of 1e−5. We checked for further ILP genes within the genome by mapping the protein sequences of these two *M. natalensis* genes and eight known *D. melanogaster* ILPs (downloaded from NCBI; accessed February 2021) against the *M. natalensis* genome. This was carried out with EXONERATE (v 2.2.0[81]) using the protein2genome model at default settings. No further ILP copies were found but the annotations of the two *M. natalensis* genes were improved based on these exonerate alignments. We searched for ILP orthologs within 25 further insect proteomes (see table in Fig. 4 for full details) using BLASTp and an e-value threshold of 1e−5. The protein sequences were aligned with t-coffee[82] in accurate mode, which incorporates both sequence profile and structural information. We trimmed the alignment with tri-mAL in the *automated1* mode[83], then created a gene tree with iqtree2[84]. This program automatically selects the best-fit model with ModelFinder and carries out bootstraps. With these methods, we recreated a gene tree Q.pfam+R4 model and 10,000 bootstraps. The tree was visualized with the online tool, iTOL v6[85].

**Metabolomic analysis**. A volume of 20 μL of hemolymph of FW, QT0, QT1, QT2, QT3, and QT4 was used to determine metabolic profiles obtained by gas chromatography coupled with mass spectrometry (GC-MS). The number of replicates per group are provided in Supplementary Table 1. We used the experimental procedure described in Khodayari et al.[86], and adapted from Genitoni et al.[87]. Samples were homogenized in 450 μL of ice-cold methanol/chloroform (2:1, v/v) before the

addition of 300 μL of ultra-pure water. After they have been vigorously vortexed, the samples were centrifuged for 10 min at 4000×*g* (4 °C). Then, 100 μL of the upper phase, which contains metabolites, was transferred to new glass vials (Thermofisher), speedvac dried at RT, and vials sealed with PTFE caps. The derivatization of the samples was conducted with a CTC CombiPAL autosampler (CTC Analytics AG, Zwingen, Switzerland), as described in Khodayari et al.[86]. The GC-MS platform consisted of an Agilent 7890B gas chromatograph coupled to a 5977B mass spectrometer. The injector was held at 250 °C, and the temperature of the oven ranged from 70 to 170 °C at 5 °C/min, from 170 to 280 °C at 7 °C/min, and from 280 to 320 °C at 15 °C/min; at the end of the temperature ramps, the oven remained at 320 °C for 4 min. A 30 m HP5 MS 30 m, I.D. 0.25 mm, thickness 0.25 μm, 5% diphenyl/95% dimethylpolysiloxane, Agilent Technologies) was used with helium as the gas carrier at 1 mL per min. The temperatures of the transfer line and ion source were 280 and 230 °C, respectively. The split mode (split ratio: 2:1) was used for the injection of 1 μL of each sample, and detection was realized by electronic impact (electron energy: 70 eV) in full-scan mode. The peaks list was annotated based on their MS fragmentation patterns with MassHunter. Detected metabolites were identified, and calibration curves were used to calculate the concentration of each metabolite.

**Lipidomic analysis**. Lipids were extracted for fatty acid profile analysis gas chromatography with flame-ionization detection (GC-FID) and gas chromatography coupled to mass spectrometry (GC-MS). Lipidomic analyses were done by liquid chromatography coupled to mass spectrometry (LC-HRMS/MS). Lipids were extracted from 20 μL of hemolymph (from FW and QT4) or from fat bodies (from QT0, QT2, and QT4) using a biphasic solvent system of cold methanol, methyl tert-butyl ether (MTBE), and water, adapted from Cajka et al.[88]. Briefly, the samples were transferred in 750 μL of MTBE and 150 μL of methanol into a 2-mL screw cap tube. For lipidomic analysis, 1 μL of internal standard (PC 31:1 | PC17:0-PC14:1) at 3775 μg/mL was added to each sample. After homogenization with the "Precellys tissue homogenizer" at 5000 rpm for 5 min, 400 μL of $H_2O$ was added to each sample. The samples were then centrifuged at 13,000 rpm for 5 min. The upper phase containing the lipids (MTBE) was transferred into a new tube and dried under a stream of nitrogen at 20 °C. For fatty acid profile analysis, extracted lipids were transferred in 100 μL of MTBE, methylated into fatty acids of methyl esters (FAMEs) after the addition of 10 μL of tetramethylammonium hydroxide (TMAH). After centrifugation at 4000 rpm for 5 min, supernatants were collected and diluted three times into heptane prior to injection into GC-FID and GC-MS. For lipidomic analysis, lipids extracted were taken up into 100 μL of isopropanol before injection into LC-HRMS/MS.

*FAMEs analysis*. From fat bodies of different queen stages, fatty acid profiles were separated and analyzed by gas chromatography with flame-ionization detection (GC-FID 2010 Plus Shimadzu) equipped with a BPX 70 capillary column (SGE, 30 m × 0.25 mm, 0.25 μm) as described in Merlier et al.[89]. The fatty acids were identified by comparison of the retention times of a standard solution of 37 fatty acid methyl esters (Sigma; 47885-U Supelco) in GC-FID and confirmed by high accuracy mass of molecular ions and their fragments after injection into a GC-MS (Q-Exactive™, Thermo)[89]. The composition of fatty acids was expressed as a relative percentage of their peak areas with respect to the total peak area of all the fatty acids. The fat body samples were normalized by dividing peak areas with total DNA concentration (ng/mL) measured with a Qubit Fluorometer (Thermofisher) and Qubit dsDNA Assay kit (Invitrogen). The membrane peroxidation index (PI) of lipid extracts in fat bodies of queens was calculated as the sum of bis-allylic methylene groups per 100 fatty acids according to the equation[67]:

$$PI = (\text{percentage of dienoics} \times 1) + (\text{percentage of trienoics} \times 2) + (\text{percentage of hexaenoics} \times 5)$$

*Untargeted lipidomics analysis*. The untargeted lipidomics analysis was conducted using a liquid chromatography-high resolution tandem mass spectrometry (LC-HRMS/MS) analysis used as described and modified from Ulmer et al.[90]. An HPLC 1290 (Agilent Technologies) coupled to a hybrid quadrupole time-of-flight high definition (QtoF) mass spectrometer Agilent 6538 (Agilent Technologies) equipped with an ESI dual-source was used. Lipids were separated on a C18 Hypersil Gold (100 × 2.1 mm, 1.9 μm, Thermofisher) at 50 °C, using an elution gradient composed of a solution of 20 mM of ammonium acetate and 0.1% formic acid (ACN:$H_2O$, 60:40, v/v) (solvent A) and a solution of 20 mM of ammonium acetate and 0.1% formic acid (IPA:ACN:$H_2O$, 90:8:2, v/v) (solvent B). Separation was conducted under the following gradient: 0–2 min from 32% (B), 2–3 min from 32 to 40% (B), 3–8 min from 40 to 45% (B), 8–10 min from 45 to 50% (B), 10–16 min from 50 to 60% (B), 16–22 min from 60 to 70% (B), 22–28 min from 70 to 80% (B), 28–30 min from 80 to 99% (B), 30–31 min from 99 to 32% (B), 31–36 min from 32 to 32% (B). The flow rate was set at 250 μL/min. Two microliters of samples were injected. MS/MS spectra were acquired in positive mode and in negative mode in data-dependent mode and MS² scans were performed on the sixth most intense ions. The source temperature, fragmentor, and the skimmer were set up at 350 °C, 150 V, and 65 V, respectively. The acquisition was made in full-scan mode between 100 *m/z* and 1700 *m/z*, with a scan of two spectra per second. Two internal references were used for in-run calibration of the mass spectrometer (121.0509, 922.0098 in positive ion mode and 112.9856, 1033.9881 in negative ion mode). MassHunter B.07 software allowed us to control the parameters of the machine acquired.

*Data processing and annotation*. MsDial v 4.0 software[91] was used for data processing and annotation of lipids. The data collection was performed by fixing the MS1 and MS2 tolerance, at 0.01 Da and 0.025 Da, respectively. The peak detection was fixed at 1000 amplitude and a mass slice width at 0.1 Da. The deconvolution parameters correspond to a sigma window value at 0.5 and a MS/MS abundance cut off at 10 amplitude. Isotopic ions were kept until 0.5 Da. The peaks list was annotated based on their unique MS/MS fragmentation patterns using the in-built LipidBlast mass spectral library in MS-DIAL software. The accurate mass tolerance MS1 and MS2 were fixed at 0.01 Da and 0.05 Da, respectively. The identification score cut off was fixed at 80%. Lipids were normalized by the intensity of the internal standard (PC 31: 1 | PC17:0-PC14:1).

**Ploidy analysis**. Fat bodies from QT0, QT1, QT2, QT3, and QT4 were processed by Flow Cytometric Analysis with a Cycletest PLUS DNA Reagent Kit (BD Biosciences, Le pont de Claix). Number of replicates per group are provided in Supplementary Table 1. All procedures were adapted from Nozaki & Matsuura[41]. Stained nuclei were analyzed for DNA-PI fluorescence using an Accuri C6 Flow Cytometer (BD Biosciences) at an excitation wavelength of 488 nm and a detector equipped with an 585/45 bandpass filter. Approximately 1000 cells were acquired for each measurement. Flow cytometric analyses were performed with the Accuri C6 software v1.0.264.21 (BD Biosciences). Debris was removed on an FSC-A/SSC-A dotplot and doublet was eliminated with PI-FL2-H/ FL2-A dotplot. The nuclei were analyzed with a histogram PI-A. The 1C DNA peak was determined by the analysis of king's testis (sperm), allowing the identification of the 2C, 4C, and 8C peaks of the others samples.

### Statistics and reproducibility

*Metabolomic analyses*. Permutational MANOVA[92] was used to test for differences in multivariate metabolite concentrations between stages and castes. We subsequently tested for differences in individual metabolites between pairs of subsequent stages or between FW vs. QT0 and FW vs. QT4. The concentrations of all metabolites were log-transformed and compared between groups by using Welch tests. Tail probabilities were corrected for multiple testing using the Benjamini–Hochberg method. Tests were considered significant for a $P$ value <0.05 and carried out using R software (v 3.6.3).

*Lipidomic analyses*. To compare the percentages of SFA, MUFA, PUFA lipid content in fat bodies of different queen stages we used a permutational MANOVA on ilr-transformed compositional data[93] followed by pairwise post hoc perMANOVA comparisons of each stage with QT4 (Holm-Bonferroni correction for multiple comparisons). To compare log-transformed individual lipid values between FW and QT4, or between QT0, QT2, and QT4, Welch tests were used, corrected for multiple testing using the Benjamini–Hochberg method. To compare the peroxidation index (PI) in fat bodies of different queen stages we used Kruskal–Wallis test followed by Dunn's post hoc comparisons. All tests were considered significant for a $P$ value <0.05 and were carried out using R software (v 4.0.2). Heatmaps were made using the heatmap function in the Metaboanalyst v 4.0. with Euclidean distance and clustering using Ward's method.

*Ploidy analyses*. To compare percentages of nuclei with different multiples of haploid genomes between queen stages, we used permutational MANOVA followed by pairwise post hoc perMANOVA (Holm–Bonferroni correction for multiple comparisons).

**Reporting summary**. Further information on research design is available in the Nature Research Reporting Summary linked to this article.

### Data availability

RNA-seq reads generated in this study have been deposited in Sequence Read Archive (BioProject ID: PRJNA685589 and BioSample accessions: SAMN17088123-SAMN17088147). The normalized expression data, differential expression results, ILP annotations, and ILP sequences are available as Supplementary Data 3–6. Further data and scripts that support the findings of this study are available in Dryad with the identifier https://doi.org/10.5061/dryad.51c59zw7t. All other data are available from the corresponding author on reasonable request.

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

## Acknowledgements

The authors would like to thank Dr. Christian Bordereau and Dr. Jannette Mitchell for helpful discussions on the termite model. The authors acknowledge the Department of Public Works and Infrastructure, Pretoria, and the University of Pretoria for allowing access to their sites of collection of *M. natalensis*. The authors also thank Cyril Fresillon and Pierre Deparscau, photographers for the CNRS images. We thank Qi Li for termite colony maintenance. This study was supported by the International Human Frontier Science Program RGP0060/2018 to M.V-C. S.S. was also supported by a fellowship from Université de Paris Est-Créteil (UPEC). A.L. acknowledges financial support from France Génomique (ANR-10-INBS-09-08). We would like to thank the EcoChim platform of the University of Rennes for access to metabolomics facilities.

## Author contributions

M.V.-C. designed the study. D.S-D., A.R., R.L., L.-A.P., Z.W.D.B., and M.V.-C. carried out termite experiments, M.V-C. and A.L. transcriptomic, D.R. and R.L. metabolomic, and S.A. and R.L. lipidomic experiments. M.A. and M.V.-C. measured DNA contents. S.S., M.C.H., T.J.M.V.D., and M.V.-C. analyzed the data, wrote the original draft, and compiled the figures and tables presented. S.S., M.C.H., D.S-D., R.L., T.J.M.V.D., A.R., L-A.P., A.L., D.R., S.A., M.A., J.V., H.S.S., Z.W.D.B., E.B-B., and M.V.-C. contributed with expertise, input, and edits throughout the text.

## Competing interests

The authors declare no competing interests.
