## [Transparent Peer Review File · Communications Biology]

Reviewers' comments:

Reviewer #1 (Remarks to the Author):

Eusocial insect such as the long-lived termite queens are good models for investigation of the tradeoff control(s) between fecundity/fertility and lifespan. The underlying mechanism still remains elusive and mysterious. In this study, the authors generated large transcriptomic, metabolomic and lipidomic datasets from the fungus-growing termite *Macrotermes natalensis*, including the short-lived workers, different stages of queens and long-lived kings. It was proposed that upregulation of an insulin-like peptide *Iip9* in the queen fat body might be essential in maintaining both high fecundity and extreme lifespan. Even it is hard to generate solid conclusions from pure omics data, the results obtained in this study may facilitate the understanding of termite fertility and longevity balancing control and provide targeted gene/metabolite for future functional investigations.

Specific comments:

First of all, the data consistency and quality are concerned but not verified or discussed by the authors. The authors' efforts are appreciated to obtain different samples of workers, multiple stages of queens, and kings. However, even it is understandably hard to obtain the isogenic lines/strains for termite animals like *M. natalensis*, it should have been realized that genetic/physiological differences can naturally exist between these termites even being collected from a relatively small area.

On top of above concerns, lipidomic analysis actually used either hemolymph (FW and QT4) or fat body samples (QT0, QT2 and QT4), which might bring additional inconsistent issues for comparative analysis. In addition, qRT-PCR analysis has not done to support the RNA-seq data, especially for the key genes such as *Ii9* and *mdy* etc. argued in this study.

Technically for RNA-seq analysis: how many nucleotide/base differences were allowed in reads mapping analysis since different samples might belong to different lines/strains of termite.

Comparative analyses of RNA-seq, metabolomic or lipidomic data have been conducted between different samples, i.e., inconsistent throughout the paper, especially with or without the "King" sample in different analysis.

The improvements for compactness are highly required: major conclusions are not well described in Abstract. The Results are presented highly descriptively, hard to follow and contains many discussions even being separated from the Discussion part.

Instead, the key conclusions/suggestions of this study have not been well discussed. For example, in contrast to the finding in this study, the expression of the upstream IIS(/TOR) signaling components was actually missing in the comparative transcriptomic analysis of termites, ants and bees in association with animal ageing and fecundity (J. Korb et al., 2021. *Philos. Trans. R. Soc. Lond. B Biol. Sci.* 376, 20190728). How to explain? In addition, it is unclear whether the juvenile hormones were detected or not in metabolomic analysis of samples in this study, which has also been suggested with an essential control in balancing insect lifespan and fecundity). Better to discuss these points.

Figures are largely huge, especially for Figures 1 and 3, very hard to follow.

Minor comments:

Where is the genome information of this termite that was referred to in this study? Gene/protein accessions are inaccessible from NCBI or UniProt. From the supplementary tables, it can be noted that, even being putative, all termite genes were subjected to *Drosophila* or human gene model. The cutoff values of similarity are unclear.

Line 471, A few months...

Reviewer #2 (Remarks to the Author):

Overall comment

This study generates three forms of 'omics' data (lipidomic, transcriptomic and metabolomic) for fat bodies derived from queens of different ages, old kings and short-lived workers. The authors try to interpret the data in the context of the longevity-fecundity paradox. Specifically, there is typically a trade off in aging and reproductive output; social insects like the termites defy this trade off in having incredibly long-lived reproductive (queens and kings in the case of termites). The authors therefore sought to use these data to identify the molecular mechanisms and signalling pathways that permit longevity. This is a neat idea, with an unprecedented dataset which would be of great value to the scientific community. However, the manuscript's message too cluttered and poorly communicated to properly understand the results, making it difficult to see the wood for the trees. Is the main message about how termite queens maximise both longevity and fecundity simultaneously? If so, trim back all the other peripherals and stick to presenting the evidence for that specific finding/hypothesis. A significant rewrite, with a critical eye for what is/isn't core to the story, and more care in expression and wording, is required to make this paper understandable and convincing. I am not providing detailed suggestions on this point, as it is the job of the authors to present their work in a coherent and concise way. My comments that follow are largely concerned with the core science, experimental design and analyses.

1. Experimental design. The question is solid, and the idea that termites of different longevity and reproductive trajectories could be used to get at the pathways of these two usually opposing processes is exciting. However, the sampling approach is not optimal. I appreciate the value of old individuals in these colonies: no-one wants to sacrifice the queen/king of 10-20 year old colonies, and they are challenging to sample. From this perspective, the authors are to be congratulated on getting hold of queens spanning the age range. But from an experimental design perspective, kings should have been sampled at similar ages – they would be expected to have the same ageing process but permit reproductive pathways to be excluded (i.e. things not shared between queens and kings). E.g. Equally, sampling workers of different ages would have been very informative. Workers don't live nearly so long and are not reproductive: one would expect workers to senesce more rapidly than queens/kings; if the TOR and IIS pathways are important in longevity in queens/kings then you'd expect these to be absent/inactive in workers. I appreciate that kings are hard to sample, but workers are relatively easy. It may not be practical to augment the sampling at this stage; and extra sampling may not be critical for this paper. However, the authors should consider how this imbalance in experimental design undermines their analyses and interpretation, and acknowledge these shortfalls.
2. Related to this, it is not clear what the pooled samples were each stage: how many samples there were for each stage, and how many individuals were pooled for each sample. There is a supplementary table that helps a little here, but overall needs better clarity on what was sequenced. Connected to this, give some specific predictions/hypotheses about what each of the comparisons test.
3. You have a great omics package: transcriptomics, metabolomics and lipidomics. This is arguably one of the most novel aspects of your study. You could make more of this: make it clear what this tri-partite analysis tells us that previous single-omics studies couldn't. But, equally, comment on the imbalance of the sampling design: e.g. the sample size of QT3 for metabolomics is only 3 compared to double for most of the other categories - why is this? How might this affect your interpretation?
4. Messaging of the study needs to be tighter especially in the Introduction. What are the big, key questions in this study? You have multiple exciting findings which should be emphasised while being

accessible to non-specialists. Too much detail in specialist information surrounding pathways etc make it a struggle to read and the ambiguous storyline in the introduction means it's hard to follow the rest of the paper. As such the introduction didn't tell a persuasive narrative that makes clear the aim of the paper. Posing some specific hypotheses would help here, and presenting the results in that form rather than descriptions of analyses. Related, can you provide a better explanation of why the termite system is a good model; related to this, are all workers female in this species? This would be clearer if your section summarising the biology of the species was moved from the methods to the introduction.

5. The flow of the Abstract could be improved; try to avoid giving a list of things you did in this paper moving into the results could be better. Jumps between background and findings a couple of times.

6. Results. Figures – too many and too complicated; too many parts. Use of word clouds for GO term enrichment in figures 1 and 3 is distracting and words are too small to be legible. Figure 1 in particular is overwhelming: too many sections and not all are that interesting/critical to the main message. Consider simplifying and/or moving to supplementary materials. Connecting the results to hypotheses/predictions that are specific to the questions posed in the introduction would help greatly. There is too much dense, less relevant detail that it's hard to know what you think is important. Ln171- confusing, did you only conduct comparisons between QT4/KT4 and FW not any of the other queens? There is not enough reference throughout about the point of these results and what they say overall, i.e., on lipid effects on ageing. The reader gets lost in all the different processes listed.

7. Discussion. As with the rest of the MS, too unfocused and rambling. The trait of having very limited storage of lipids in favour of immediate utilisation seems to be something that is not achievable under most life strategies, and is quite specific to the lifestyle of a queen, who does not have to engage in food acquisition and is constantly supplied food by her workers – might it be important to highlight that this may be why the trade-off between fecundity and longevity is generally so ubiquitous, but is avoided in this system.

Reviewer #3 (Remarks to the Author):

This is a big "omics" study revealing how a natural system has modified gene expression on a shared genetic background to achieve long lifespans. Differences in gene expression, the metabolome, lipidome, and ploidy were all examined. The study conclusively showed how this natural system adapted to overcome the proposed causes of aging by at least three of the mainstream molecular theories for aging, mitochondria theory, the free radical theory, and damage accumulation theory. By showing changes in gene expression and the metabolome consistent with overcoming each of popular explanations for aging the study proves all three are correct and are not mutually exclusive. The authors undersold the significance of their findings that both supports all three, and that none of the three are the sole correct answer to aging. The proposed and favored hypothesis is that a new ILP they discovered in termites and some other insects changes TOR signaling to switch on long-lived gene expression profiles for the other mechanisms to extend lifespan. The study consists of a very large body of convincing work and is very well written. There are a couple of problems that must be addressed before publication, and I have a few suggestions on points that could be more emphasized.

Problems that need to be addressed:

1) I will start off by saying that sample set up and statistical analysis is extremely difficult and complicated with genomic studies of social insects. In the text, it was not explained well enough to follow that a full set was sampled from each of 6 colonies as obvious in Figure 1b. But the species identification section mentions 9 colonies? It is also unclear what is being pooled, individuals from the

same colony and/or across colonies by age cohort. Is each study done with the same set of colonies sampled the same number of times? The actual mapping of the colony level sampling needs to be clarified in the text across all experiments, and ideally in supplemental table 9. See the confusion that I note below on lines 131 and 531. I simply could not untangle all of this.

2)The phylogenetic analysis (lines 598 -613 and figure S5) is not anywhere near ready for publication and either needs to be struck or redone. It looks like a quick token analysis to make figure S5. Critical information was left out. For example, which distance model was used or is Supp fig. 5 or is it a maximum parsimony tree? The sequence identifiers are not all standard in that simply searching them in some cases will not even yield one hit at Genbank or any other database I tried. If the authors need to rename or modify identifiers, then a table in the supplemental section mapping the aliases is required.

The figure S5 has no confidence limits or even bootstraps shown for branch support, and the legend needs to specify that it is a protein sequence tree.

It is not that much more work to do this part correctly. The investigators just need to re-examine the data file using something like MEGAX starting with choosing the best model then building trees with bootstrapping or some other measure of branch reliability. They could also do a parsimony analysis or some other tree construction method, but whichever way they do it, the program parameters need to be shown and the choices for those parameters justified. Finally, the aligned data set needs to be made available, ideally in the PopSet section of GenBank. GitHub is fine for public archiving, but the work will be more noticed in the major sequence repositories.

Suggestions:

1)The most unexpected finding for me is that trehalose is negatively correlated with long life. This is opposite the findings in mammals and even in *C. elegans*. This speaks to the nonpublic nature of how aging evolves. The manuscript would benefit from more discussion of this interesting contrast. (put into paragraph starting on line 357)

2)The discussion needs to be more "flashy" or aggressive in stating how multiple problems that that have been previously linked to aging are solved by social insects. This not only validates these theories from another angle, but also speaks to the near impossibility of fining a silver bullet treatment or cure for aging in humans. The really big discovery here is that not one thing changed, but that four of what many treat as independent causes of aging have been changed by evolution to lengthen lifespan. There are probably thousands of supplements and treatments being suggested to cure human aging based on solving the problems caused by any one of those four theories (insulin, mitochondrial, free radical, and damage accumulation). Here we learn from a natural system that a more integrated approach solving all four at once is going to be necessary to extend lifespan. The fact is humans probably do not have anything like the ILP9 and the eIF6-crc pathway that can make all of the necessary modifications. This study debunks the idea of a magic bullet in humans unless it can cause similar global metabolic changes in all of the areas above. That could be the popular press "hook" and it should go in the abstract as well.

3) Not necessarily for this publication, but the pathways turned on by ILP9 and the eIF6-crc are the general type that are regulated by histone acetylation and DNA methylation in other organisms. Any signs of these types of regulation in termites here?

The details (by line):

43) Oxford comma after long-lived kings

131 -132) Unclear what the four groups are. It looks like only three are listed?

135) is it "up or down regulated"? Double check this

147) the first sentence needs a citation for a review on termites or to Methods where there is a very good summary of the life history.

148-149) move the figure reference to the end of the sentence.

543) What does 4 FW, 4 QT1 etc. mean? 1 FW from 4 different colonies or 4 workers from one colony? Again, the actual sampling regime is not explained well enough. This might require a figure in addition to supplementary table 9.

604) Capitalize Exonerate

1070 -1072) there seem to be some extra spaces that need to be struck.

In conclusion:

This a very large and well done study showing how nature has solved the major mechanisms previously found to cause aging. However there are two major but fixable flaws that need to be fixed before publication. First the sample pooling with respect to colony sampling needs to be clarified as it is unintelligible as currently written, and second the phylogenetic analysis needs to be raised to the same high standard as the rest of the analyses in the paper. After those two fixes, and possibly elaborating a bit more on trehalose and aging in other organisms and bringing more forward the significance of the multifactorial fixes that termites have in place to live longer, I think this will be an outstanding contribution to the fields.

Please find below our responses (in red) to each reviewer comment.

Reviewers' comments: Reviewer #1 (Remarks to the Author):

Eusocial insect such as the long-lived termite queens are good models for investigation of the tradeoff control(s) between fecundity/fertility and lifespan. The underlying mechanism still remains elusive and mysterious. In this study, the authors generated large transcriptomic, metabolomic and lipidomic datasets from the fungus-growing termite *Macrotermes natalensis*, including the short-lived workers, different stages of queens and long-lived kings. It was proposed that upregulation of an insulin-like peptide Ilp9 in the queen fat body might be essential in maintaining both high fecundity and extreme lifespan. Even it is hard to generate solid conclusions from pure omics data, the results obtained in this study may facilitate the understanding of termite fertility and longevity balancing control and provide targeted gene/metabolite for future functional investigations.

Author reply: We appreciate the generally positive appreciation of our manuscript. Below we reply to each specific comment.

Specific comments:

1-First of all, the data consistency and quality are concerned but not verified or discussed by the authors. The authors' efforts are appreciated to obtain different samples of workers, multiple stages of queens, and kings. However, even it is understandably hard to obtain the isogenic lines/strains for termite animals like *M. natalensis*, it should have been realized that genetic/physiological differences can naturally exist between these termites even being collected from a relatively small area.

Author reply: We collected replicates of each analyzed caste and queen stage from separate, independent colonies (see the new Supplementary Tables 1 and 9) which could indeed be genetically different next to genetic differences between kings and queens within a colony and segregation variance among workers. We state now that lifespan differences between castes within colonies depend on phenotypic plasticity (Lines 68 and following, Line 82, Line 427) and note that our study of queen stages can be interpreted as an investigation of developmental plasticity (Line 96, Line 126 and following). Genetic variation can augment the variances between replicates *within* castes and queen stages, and affect comparisons *between* castes and queen stages (genotype by environment interactions). We ensured replicates of each caste and queen stage from at least three independent colonies in each comparison of caste- and stage-specific effects in gene expression. The PCA (previous Fig. 1B, now Supplementary Fig. 1) clearly shows the relevance of this replication in demonstrating that potential genetic differences in plasticity are limited. In all cases, differences between individuals from different colonies are smaller than between castes and queen stages.

2-On top of above concerns, lipidomic analysis actually used either hemolymph (FW and QT4) or fat body samples (QT0, QT2 and QT4), which might bring additional inconsistent issues for comparative analysis.

Author reply: Hemolymph and fat body lipidomic analyses provide complementary information and inconsistencies are in fact informative. Hemolymph composition integrates sources and sinks from multiple tissues and therefore informs on which resources are available globally within the body. Fat body analysis reveals what is produced/used by the fat body and the comparison with differential gene expression analysis renders our conclusions on that organ more powerful. We note that our analysis is not comparative in the usual biological

sense, we don't compare different species. We compare castes and stages to understand phenotypic plasticity prolonging lifespan.

In addition, qRT-PCR analysis has not been done to support the RNA-seq data, especially for the key genes such as *Il9* and *mdy* etc. argued in this study.

Author reply: We have performed a robust RNAseq analysis which incorporates at least three replicates per caste or queen stage, allowing us to quantify expression levels across all protein coding genes. We have used well-established methods (*hisat2*, *htseq*, *DESeq2*) that can robustly quantify relative levels of gene expression, while controlling and normalizing for technical variation between samples. We had considered performing qPCR studies to 're-validate' some of our gene-expression findings but there is little evidence that qPCR analyses from the same samples will add any extra utility to our data so we decided to eschew those experiments. Previous studies have shown extremely close correlations between qPCR and RNAseq data (*Asmann et al.*, 2009; *Griffith et al.*, 2010; *Shi and He*, 2014; *Wu et al.*, 2014). Ideally, we would re-validate our findings (potentially by qPCR) in a separate cohort of samples, but due to the difficulty of obtaining these additional samples in this period of COVID, those experiments are not possible at this time.

Asmann YW, Klee EW, Thompson EA, Perez E a, Middha S, et al. 3' tag digital gene expression profiling of human brain and universal reference RNA using Illumina Genome Analyzer. *BMC Genomics* 10: 531(2009).

Griffith M, Griffith OL, Mwenifumbo J, Goya R, Morrissy a S, et al. Alternative expression analysis by RNA sequencing. *Nat Methods* 7: 843–847 (2010).

Shi Y, He M Differential gene expression identified by RNA-Seq and qPCR in two sizes of pearl oyster (*Pinctada fucata*). *Gene* 538 : 313–322 (2014).

Wu AR, Neff NF, Kalisky T, Dalerba P, Treutlein B, et al. Quantitative assessment of single-cell RNA-sequencing methods. *Nat Methods* 11: 41–46 (2014).

Technically for RNA-seq analysis: how many nucleotide/base differences were allowed in reads mapping analysis since different samples might belong to different lines/strains of termite.

Author reply: We mapped reads to the genome with *hisat2* at default settings (see Methods section Lines 524 and following). This program sets the minimum score of an alignment based on read length following the function: $f(x) = 0 + -0.2 * x$; where x is the read length. We intentionally worked with replicates of each caste and queen stage from different colonies, in order to control for relatedness and obtain more systematic effects of caste or queen stage on gene expression. The reads of all samples were mapped to the same reference genome (*Poulsen et al.*, 2014), so that we expected no biases in mapping rates caused by differences in genotypes. Even if differences in overall mapping rates could have been linked to genotype, i.e. colony of origin, these would have been evened out when normalizing counts (see manual of *DESeq2*) and, in any case, should not cause biases among gene loci. Nevertheless, we have rechecked rates of unambiguously mapped reads from all RNAseq libraries. These ranged from 54.14-83.13%, with a mean of 70.0-77.0% per colony. An ANOVA reveals no significant effect of colony ($F(5,19) = 0.34, p = 0.88$) or caste ($F(6,18) = 1.95, p = 0.13$) on mapping rate.

Poulsen, M. et al. Complementary symbiont contributions to plant decomposition in a fungus-farming termite. *PNAS* 111, 14500–14505 (2014).

Comparative analyses of RNA-seq, metabolomic or lipidomic data have been conducted between different samples, i.e., inconsistent throughout the paper, especially with or without the “King” sample in different analysis.

Author reply: The focus of our study was on queens because the largest maturation changes were observed there. We constrained our longitudinal analysis to queen stages because there are fewer predictions of developmental plasticity in kings and because of practical limitations.

3-The improvements for compactness are highly required: major conclusions are not well described in Abstract. The Results are presented highly descriptively, hard to follow and contains many discussions even being separated from the Discussion part.

Author reply: We have rewritten the abstract with clearer major conclusions (Line 43 on lifespan, Lines 47-49 on the absence of a trade-off) and changed the title accordingly. We have reworked the Results and Discussion sections to make them less descriptive, and moved the many discussions into the actual Discussion section.

Instead, the key conclusions/suggestions of this study have not been well discussed. For example, in contrast to the finding in this study, the expression of the upstream IIS(/TOR) signaling components was actually missing in the comparative transcriptomic analysis of termites, ants and bees in association with animal ageing and fecundity (J. Korb et al., 2021. *Philos. Trans. R. Soc. Lond. B Biol. Sci.* 376, 20190728). How to explain? In addition, it is unclear whether the juvenile hormones were detected or not in metabolomic analysis of samples in this study, which has also been suggested with an essential control in balancing insect lifespan and fecundity). Better to discuss these points.

Author reply: In this revised version of the manuscript we now draw attention to these points regarding IIS/TOR pathways in the discussion (Lines 378 and following). We studied the fat body while most recent studies in eusocial insects addressing lifespan and the longevity-fecundity trade-off used whole body analysis or more recently, analyzed heads and thorax. The abdominal fat body was not investigated except in a single ant model (Negroni *et al.* 2019). The abdominal fat body has specific metabolic functions and results therefore don't need to match those from other tissues and organs. Concerning the juvenile hormone, its measurement would have required a different specific methodology and further sampling. We were not able to execute it at this time because the small amounts of material collected were used to determine metabolites and lipids, not hormones. If we had chosen otherwise, it would have added more insight in the functioning of the IIS/TOR pathways at the detriment of understanding what happens with the different types of lipids and membrane peroxidability. These are "closer" to actual lifespan affecting processes and we therefore chose to analyze those.

Negroni, M. A., Foitzik, S., & Feldmeyer, B. Long-lived *Temnothorax* ant queens switch from investment in immunity to antioxidant production with age. *Scientific reports*, 9(1), 1-10(2019).

4- Figures are largely huge, especially for Figures 1 and 3, very hard to follow.

Author reply: All figures and their captions have been improved. Especially Fig. 1 has been simplified by splitting into several separate figures (Fig. 1 and Supplementary Figs. 1 and 2).

Minor comments:

5- Where is the genome information of this termite that was referred to in this study? Gene/protein accessions are inaccessible from NCBI or UniProt. From the supplementary tables, it can be noted that, even being putative, all termite genes were subjected to *Drosophila* or human gene model. The cutoff values of similarity are unclear.

Author reply: We cite the original article, which published the *Macrotermes natalensis* genome assembly (Poulsen *et al.*, 2014). That publication links to the genomic resources: <http://gigadb.org/dataset/100057> \o "http://gigadb.org/dataset/100057. We have added this information in the methods (Line 523).

Orthology to *Drosophila* and *Human* genes was determined with the method of reciprocal best blast hit. We realized that this was omitted from the methods and have now added a section to cover this topic (Line 553 and following). We have also added two files to the github repository which contain the gene IDs and e-values of these orthologous pairs (ortholog_pairs_Dmel_Mnat.txt & ortholog_pairs_Hsap_Mnat.txt).

Poulsen, M. et al. Complementary symbiont contributions to plant decomposition in a fungus-farming termite. *PNAS* 111, 14500–14505 (2014).

6- Line 471, A few months...

Author reply: Thank you for pointing this out, we have corrected this (Line 461)

Reviewer #2 (Remarks to the Author):

Overall comment. This study generates three forms of ‘omics’ data (lipidomic, transcriptomic and metabolomic) for fat bodies derived from queens of different ages, old kings and short-lived workers. The authors try to interpret the data in the context of the longevity-fecundity paradox. Specifically, there is typically a trade off in aging and reproductive output; social insects like the termites defy this trade off in having incredibly long-lived reproductive (queens and kings in the case of termites). The authors therefore sought to use these data to identify the molecular mechanisms and signalling pathways that permit longevity. This is a neat idea, with an unprecedented dataset which would be of great value to the scientific community. However, the manuscript’s message too cluttered and poorly communicated to properly understand the results, making it difficult to see the wood for the trees. Is the main message about how termite queens maximise both longevity and fecundity simultaneously? If so, trim back all the other peripherals and stick to presenting the evidence for that specific finding/hypothesis. A significant rewrite, with a critical eye for what is/isn’t core to the story, and more care in expression and wording, is required to make this paper understandable and convincing. I am not providing detailed suggestions on this point, as it is the job of the authors to present their work in a coherent and concise way. My comments that follow are largely concerned with the core science, experimental design and analyses.

Author reply: Thank you for your positive words. Our initial main focus of the study was indeed on how termite queens maximize both longevity and fecundity. Since then, and as also suggested by the summary of our research

question by this reviewer, we realized that we actually address which molecular mechanisms and signaling pathways permit longevity in termite reproductives. In the case of termite queens, this requires that we additionally understand how they sustain a massive reproductive effort without decreasing lifespan. Our main message is that plastic changes in several molecular mechanisms permit longevity, and that queen oogenesis occurs without detrimental side effects such as unnecessary fat storage. We have now rewritten large parts of our manuscript to better present this main message better and also adjusted title and abstract. We have carefully reworded out text as detailed in our replies to several comments by all three reviewers.

1-Experimental design. The question is solid, and the idea that termites of different longevity and reproductive trajectories could be used to get at the pathways of these two usually opposing processes is exciting. However, the sampling approach is not optimal. I appreciate the value of old individuals in these colonies: no-one wants to sacrifice the queen/king of 10-20-year-old colonies, and they are challenging to sample. From this perspective, the authors are to be congratulated on getting hold of queens spanning the age range. But from an experimental design perspective, kings should have been sampled at similar ages – they would be expected to have the same ageing process but permit reproductive pathways to be excluded (i.e. things not shared between queens and kings). E.g. Equally, sampling workers of different ages would have been very informative. Workers don't live nearly so long and are not reproductive: one would expect workers to senesce more rapidly than queens/kings; if the TOR and IIS pathways are important in longevity in queens/kings then you'd expect these to be absent/inactive in workers. I appreciate that kings are hard to sample, but workers are relatively easy. It may not be practical to augment the sampling at this stage; and extra sampling may not be critical for this paper. However, the authors should consider how this imbalance in experimental design undermines their analyses and interpretation, and acknowledge these shortfalls.

Author reply: We agree on this fully. For the longitudinal part of this study we decided to focus on queen development. Please just consider the total amount of comparisons and more stringent corrections to control false positives to make in a fully balanced longitudinal design with three castes of different ages. In our opinion, it also requires additional preparatory work to remedy a number of unknowns, for example to determine the expected changes in maturation stages and reproductive investment in kings with age. Such data were not available and expectations could not be formulated, in contrast to what we already knew on queen maturation (starvation, physogastry, change in fat body identity and function). Nevertheless, we also agree that future work on aging in kings and workers of this termite would be very valuable for putting our new findings on queen maturation into perspective. We now acknowledge some of these aspects by pointing in the discussion to follow-up work on kings (Lines 352 and following, Line 370 and following, Line 424 and following).

2- Related to this, it is not clear what the pooled samples were each stage: how many samples there were for each stage, and how many individuals were pooled for each sample. There is a supplementary table that helps a little here, but overall needs better clarity on what was sequenced. Connected to this, give some specific predictions/hypotheses about what each of the comparisons test.

Author reply: We now supply a new Supplementary Table 1, which clearly outlines our sampling strategy for each experiment. In order to clarify the choice of comparisons performed in this paper we mention our expectations and tested hypotheses in the introduction (Lines 94 and following plus Lines 101 and following).

3-. You have a great omics package: transcriptomics, metabolomics and lipidomics. This is arguably one of the most novel aspects of your study. You could make more of this: make it clear what this tri-partite analysis tells

us that previous single-omics studies couldn't. But, equally, comment on the imbalance of the sampling design: e.g. the sample size of QT3 for metabolomics is only 3 compared to double for most of the other categories - why is this? How might this affect your interpretation?

Author reply: Thank you for your positive words. We now mention in the introduction (Lines 89 and following) the tri-partite analysis. Concerning the sample size of QT3, the number of samples collected at T1 and T2 and mortality among the other colonies reduced the number of individuals available for sampling at T3. As far as we know, nobody else has yet succeeded in obtaining sufficient colonies of 31-month-old animals (QT3) for a molecular and physiological analysis. We agree that a larger sample size at QT3 would have been better, as for QT4, to potentially detect additional significant differences. However, Fig. 6 shows that the discriminant analysis managed to separate QT3 just as well from the other stages. Also, for a given variance of a response, doubling the sampling size for QT3 would have reduced the standard error of estimates on QT3 by a factor of only 0.7, approximately, which is a limited reduction.

4- Messaging of the study needs to be tighter especially in the Introduction. What are the big, key questions in this study? You have multiple exciting findings which should be emphasised while being accessible to non-specialists. Too much detail in specialist information surrounding pathways etc make it a struggle to read and the ambiguous storyline in the introduction means it's hard to follow the rest of the paper. As such the introduction didn't tell a persuasive narrative that makes clear the aim of the paper. Posing some specific hypotheses would help here, and presenting the results in that form rather than descriptions of analyses. Related, can you provide a better explanation of why the termite system is a good model; related to this, are all workers female in this species? This would be clearer if your section summarizing the biology of the species was moved from the methods to the introduction.

Author reply: We have reduced the detailed pathway information in the introduction as much as possible. We pose clear hypotheses, also in response to the remarks by the other two reviewers. We have reworked the introduction and use elements from the paragraph on termite biology to showcase their interest and to argue why they are a good model in the introduction (Lines 67 and following). Two types of workers exist in this species (Lines 69 and following), and we mention that we collected minor (female) workers on Lines 125 and 439. We have also restructured the results around our specific questions.

5- The flow of the Abstract could be improved; try to avoid giving a list of things you did in this paper moving into the results could be better. Jumps between background and findings a couple of times.

Author reply: We have changed the abstract. Background and findings are more clearly kept separate. We avoid lists of things when they are not essential.

6- Results. Figures – too many and too complicated; too many parts. Use of word clouds for GO term enrichment in figures 1 and 3 is distracting and words are too small to be legible. Figure 1 in particular is overwhelming: too many sections and not all are that interesting/critical to the main message. Consider simplifying and/or moving to supplementary materials. Connecting the results to hypotheses/predictions that are specific to the questions posed in the introduction would help greatly. There is too much dense, less relevant detail that it's hard to know what you think is important. Ln171- confusing, did you only conduct comparisons between QT4/KT4 and FW not any of the other queens? There is not enough reference throughout about the point of these results and what they say overall, i.e., on lipid effects on ageing. The reader gets lost in all the different processes listed.

Author reply: We have separated Fig. 1 into several separate images and moved less important information to the supplementary materials. GO tag clouds have now been replaced by a list of the top 3 GO terms per module in the new Fig. 2, while the detailed GO-enrichment results have been now presented in supplementary Table 2. We explicitly connect results to specific predictions/expectations and tried to remove less relevant details as much as possible as well as lists of processes.

The comment on our previous Line 171: We did not just compare QT4/KT4 with FW as our text evidences, but we wanted to focus in the paragraph in question on expression changes shared by reproductives of both sexes relative to female minor workers. The paragraph has been rewritten to make this clearer (Lines 153 and following).

7- Discussion. As with the rest of the MS, too unfocused and rambling. The trait of having very limited storage of lipids in favor of immediate utilization seems to be something that is not achievable under most life strategies, and is quite specific to the lifestyle of a queen, who does not have to engage in food acquisition and is constantly supplied food by her workers – might it be important to highlight that this may be why the trade-off between fecundity and longevity is generally so ubiquitous, but is avoided in this system.

Author reply: We have tried to focus the discussion based on remarks given by all three reviewers, and now address lifespan affecting processes in turn, in an order that helps to avoid redundancies. We have emphasized the point that a termite queen avoids costs of food acquisition (Line 358 and following).

Reviewer #3 (Remarks to the Author):

This is a big “omics” study revealing how a natural system has modified gene expression on a shared genetic background to achieve long lifespans. Differences in gene expression, the metabolome, lipidome, and ploidy were all examined. The study conclusively showed how this natural system adapted to overcome the proposed causes of aging by at least three of the mainstream molecular theories for aging, mitochondria theory, the free radical theory, and damage accumulation theory. By showing changes in gene expression and the metabolome consistent with overcoming each of popular explanations for aging the study proves all three are correct and are not mutually exclusive. The authors undersold the significance of their findings that both supports all three, and that none of the three are the sole correct answer to aging. The proposed and favored hypothesis is that a new ILP they discovered in termites and some other insect changes TOR signaling to switch on long-lived gene expression profiles for the other mechanisms to extend lifespan. The study consists of a very large body of convincing work and is very well written. There are a couple of problems that must be addressed before publication, and I have a few suggestions on points that could be more emphasized.

Author reply: Thank you for your positive words and for your helpful comments. We now stress that there is no single correct answer to the question of what permits long lifespan. It is one of our major conclusions and we restructured our text to bring this more to the foreground (Lines 432 and following).

Problems that need to be addressed

1-I will start off by saying that sample set up and statistical analysis is extremely difficult and complicated with genomic studies of social insects. In the text, it was not explained well enough to follow that a full set was sampled from each of 6 colonies as obvious in Figure 1b. But the species identification section mentions 9

colonies? It is also unclear what is being pooled, individuals from the same colony and/or across colonies by age cohort. Is each study done with the same set of colonies sampled the same number of times? The actual mapping of the colony level sampling needs to be clarified in the text across all experiments, and ideally in supplemental table 9. See the confusion that I note below on lines 131 and 531. I simply could not untangle all of this.

Author reply: We agree that sampling was not sufficiently well described in the original manuscript. We have now produced a Supplementary Table 1 that clearly outlines our sampling strategy for each experiment (transcriptomic, metabolomic and lipidomic). We have clarified this in the text (for example on Lines 123 and following) and across all experiments.

2-The phylogenetic analysis (lines 598 -613 and figure S5) is not anywhere near ready for publication and either needs to be struck or redone. It looks like a quick token analysis to make figure S5. Critical information was left out. For example, which distance model was used or is Supp fig. 5 or is it a maximum parsimony tree? The sequence identifiers are not all standard in that simply searching them in some cases will not even yield one hit at Genbank or any other database I tried. If the authors need to rename or modify identifiers, then a table in the supplemental section mapping the aliases is required. The figure S5 has no confidence limits or even bootstraps shown for branch support, and the legend needs to specify that it is a protein sequence tree. It is not that much more work to do this part correctly. The investigators just need to re-examine the data file using something like MEGAX starting with choosing the best model then building trees with bootstrapping or some other measure of branch reliability. They could also do a parsimony analysis or some other tree construction method, but whichever way they do it, the program parameters need to be shown and the choices for those parameters justified. Finally, the aligned data set needs to be made available, ideally in the PopSet section of GenBank. GitHub is fine for public archiving, but the work will be more noticed in the major sequence repositories.

Author reply: This analysis has been thoroughly redone. First, we added genes from a larger set of insect species including additional Blattodea species. The protein sequences were then aligned with t-coffee in the accurate mode which incorporates both sequence profile and structural information to arrive at as accurate an alignment as possible. The tree was then created with iqtree2. This program automatically selects the best-fit model with ModelFinder and carries out bootstraps. With these methods we recreated a gene tree with Q. pfam+R4 model and 10,000 bootstraps. The tree is now more accurate and contains valuable bootstrap information, while also reconfirming our previous findings. We have added this to the methods on Lines 571 and following.

Suggestions:

3-The most unexpected finding for me is that trehalose is negatively correlated with long life. This is opposite the findings in mammals and even in *C. elegans*. This speaks to the nonpublic nature of how aging evolves. The manuscript would benefit from more discussion of this interesting contrast. (put into paragraph starting on line 357).

Author reply: This is a crucial point showing the importance of carrying out metabolomics next to differential gene expression. We refer to the mentioned results in other model organisms and bring this point more to the foreground in the discussion (Lines 344 and following).

4) The discussion needs to be more “flashy” or aggressive in stating how multiple problems that that have been previously linked to aging are solved by social insects. This not only validates these theories from another angle,

but also speaks to the near impossibility of finding a silver bullet treatment or cure for aging in humans. The really big discovery here is that not one thing changed, but that four of what many treat as independent causes of aging have been changed by evolution to lengthen lifespan. There are probably thousands of supplements and treatments being suggested to cure human aging based on solving the problems caused by any one of those four theories (insulin, mitochondrial, free radical, and damage accumulation). Here we learn from a natural system that a more integrated approach solving all four at once is going to be necessary to extend lifespan. The fact is humans probably do not have anything like the ILP9 and the eIF6-crc pathway that can make all of the necessary modifications. This study debunks the idea of a magic bullet in humans unless it can cause similar global metabolic changes in all of the areas above. That could be the popular press “hook” and it should go in the abstract as well.

Author reply: Again, thank you for these helpful comments. We rewrote the abstract and we have revised the conclusions in light of your comments. Please check our conclusion on Lines 427 and following. We do note the peculiar situation that termite reproductives don't need to search for food or store reserves (Lines 358 and following). That "function" is dealt with by the worker castes and the cultivation of fungi. This might have made it easier to make all evolutionary changes in phenotypic plasticity, because there is a constraint less. It would then also imply that in humans, constraints on removing aging might remain present and we might never be able to solve all four challenges to aging at once. We have decided to change our title based on these remarks and believe that it now provided a more popular "hook".

5) Not necessarily for this publication, but the pathways turned on by ILP9 and the eIF6-crc are the general type that are regulated by histone acetylation and DNA methylation in other organisms. Any signs of these types of regulation in termites here?

Author reply: Thank you for this remark and question. We have submitted a manuscript elsewhere investigating the complex regulatory role of DNA methylation on caste-specific gene expression in this eusocial termite.

The details (by line):

43) Oxford comma after long-lived kings

Author reply: Thanks, we corrected this.

131 -132) Unclear what the four groups are. It looks like only three are listed?

Author reply: We deleted this sentence.

135) is it “up or down regulated”? Double check this

Author reply: The expression of some modules was either up- or down-regulated (negatively or positively correlated) in individual castes or queen stages. We realize that our previous Fig. 1 was rather confusing but we believe that our new Fig. 2 and supplementary Fig. 2 are much more intuitive. For example, the red module is upregulated in young queens (QT0 & QT2), while the pink module is up-regulated in QT2 but downregulated in QT4.

147) the first sentence needs a citation for a review on termites or to Methods where there is a very good summary of the life history.

Author reply: We have reworked this subsection and integrated it into the discussion. A more general citation has been added.

148-149) move the figure reference to the end of the sentence

Author reply: We have moved the figure reference to the end of the sentence (Line 156).

543) What does 4 FW, 4 QT1 etc. mean? 1 FW from 4 different colonies or 4 workers from one colony? Again, the actual sampling regime is not explained well enough. This might require a figure in addition to supplementary table 9.

Author reply: This means, for example, workers from four colonies, KT4 from three colonies, etc. We have deleted this sentence and replaced by a new Supplementary Table 1 which clearly outlines our sampling strategy for each experiment (transcriptomic, metabolomic and lipidomic) including colony of origin, numbers of replicates and sample pooling for analysis.

604) Capitalize Exonerate

Author reply: Done (Line 565)

1070 -1072) there seem to be some extra spaces that need to be struck.

Author reply: This was caused by block alignment combined with very long words.

In conclusion: This a very large and well-done study showing how nature has solved the major mechanisms previously found to cause aging. However, there are two major but fixable flaws that need to be fixed before publication. First the sample pooling with respect to colony sampling needs to be clarified as it is unintelligible as currently written, and second the phylogenetic analysis needs to be raised to the same high standard as the rest of the analyses in the paper. After those two fixes, and possibly elaborating a bit more on trehalose and aging in other organisms and bringing more forward the significance of the multifactorial fixes that termites have in place to live longer, I think this will be an outstanding contribution to the fields.

Author reply: We greatly appreciate these very positive comments and helpful suggestions for improvement. We have endeavored to implement these suggestions, especially regarding our sampling strategy (see Supplementary Table 1), the phylogenetic analysis of the ILP genes (see Fig. 4 and Line 560 and following) and discussing the implications of trehalose on aging (see Lines 344 and following). Title, Abstract and main text have all been adapted to convey the result that there are multifactorial fixes required and present.

REVIEWERS' COMMENTS:

Reviewer #1 (Remarks to the Author):

The authors have done substantial improvements in this revision by re-writings, re-structuring and refiguring etc. It is now straightforwardly readable. My concerns have also been answered, even arguably. I would like to recommend a suggestion for acceptance for this revision.

Reviewer #3 (Remarks to the Author):

It looks like you have responded adequately to all of my concerns. The phylogenetic analysis in particular looks much better and the sampling is clearly explained now. I think it should be accepted and published at this point.